# Low-Rank Interconnected Adaptation across Layers

## Abstract

Low-rank adaptation (LoRA) is a powerful parameter-efficient fine-tuning method that utilizes low-rank projectors $A$ and $B$ to learn weight updates $\Delta W$ for adaptation targets $W$. However, while the low-rank structure of $A$ and $B$ enables high hardware efficiency, it also restricts the overall weight update to be low-rank, which limits the adaptation performance. In this paper, we propose low-rank interconnected adaptation across layers (Lily). Specifically, we employ a hierarchical framework where low-dimensional projectors (LPs) retained for downward projection at a particular level, while globally-shared high-dimensional projector (HP) experts perform upward projection across all levels of layers. This interconnected asymmetric structure makes the adaptation much more dynamic and breaks the low-rank weight-update constraint of LoRA when using the same parameters budget. Furthermore, Lily's cross-layer connections facilitate the capture of intricate information and dependencies across different layers, thereby enhancing the model's representational capabilities. Experiments across various modalities, architectures, and model sizes underscore Lily's great performance and efficiency.

## 1 Introduction

For foundation models like Transformers (Vaswani et al., 2017b), fine-tuning on downstream tasks is a typical usage, but full fine-tuning (FFT) of large models like large language models (LLMs) incurs huge computational and storage costs and risks forgetting previously learned knowledge (Biderman et al., 2024). Linear probing, which fine-tunes only the final modules like classification heads, addresses these issues but leads to significant performance degradation since it doesn't update weights from the backbone. To tackle these challenges, parameter-efficient fine-tuning (PEFT) has received significant attention. In PEFT, a model's backbone weights are frozen, and lightweight trainable modules are introduced to efficiently learn task-specific knowledge. Among all PEFT methods, Low-rank Adaptation (LoRA (Hu et al., 2021)) is one of the most widely applied techniques, especially in LLMs. LoRA introduces a pair of low-rank projection matrices for each adaptation target, consisting of a downward adapter $A$ and an upward adapter $B$, to approximate $\Delta W$ in FFT. Due to its low-rank nature, LoRA offers significant computational and storage savings, effectively alleviating the burdens of FFT while significantly outperforming linear probing by learning the weight updates for backbone weight.

However, LoRA and many subsequent improvements to the method (Miles et al., 2024), (Zhang et al., 2023), (Zhong et al., 2024) have a limitation: the overall learned weight updates $\Delta W$ are also restricted to be low-rank because of its low-rank structure, which limits the model performance during adaptation. We recognize that one of the problems lies in the fact that the source of information is limited for each adaptation target in LoRA, as shown in Fig. 1. It can be observed that each layer in LoRA receives information only from the very layer they are situated. This prompts a question: *How can we enable a more dynamic and expressive adaptation with high-rank weight-updates by providing more sources of information for an adaptation target?*

In this paper, we propose **L**ow-rank **i**nterconnected adaptation across **l**a**y**ers (Lily), a novel framework for more expressive and performative PEFT. Specifically, we decouple the downward low-dimensional projector (LP) and its corresponding upward high-dimensional projectors (HP), making them not tightly-bonded. Each LP is connected to all the HPs, and vice versa, as shown in Fig. 1. This results in a hierarchical structure where LPs are still retained at a particular level to perform

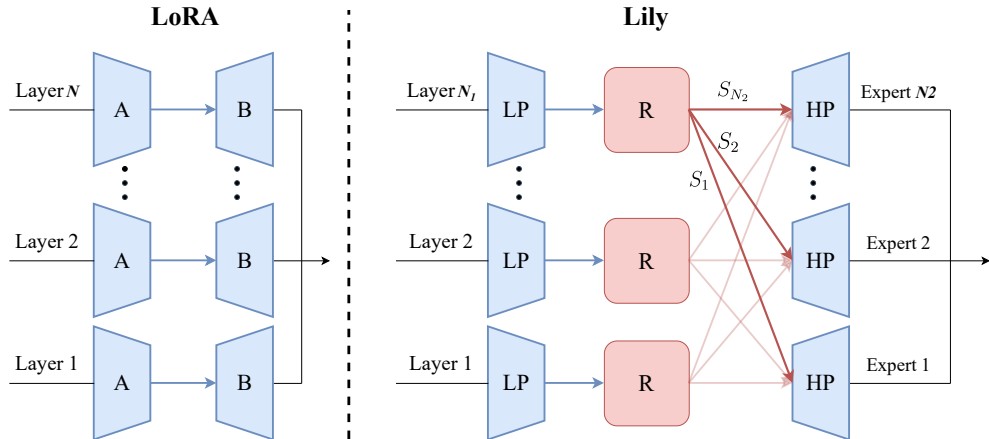

Figure 1: Dynamics of LoRA and Lily. $N$ is used to denote the number of layers in the model in the LoRA setup. Meanwhile, $N_1$ denotes the number of low-dimensional projectors and $N_2$ denotes the number of high-dimensional projector experts. R is representing the router from Lily. $N_1$ and $N_2$ can be flexibly set, independent of the number of layers.

downward projection, while all HPs are now globally shared by all the LPs, performing upward projection. Inspired by self-attention Vaswani et al. (2017a), which calculates the relationship between a token and all tokens and obtains attention scores indicating the strength of their relationship, we selectively connect an LP with the HPs based on layer features. The LP extracts features from the current layer, and based on the extracted features, a data-dependent and selective combination of HPs is performed. This is realized by utilizing a router (Shazeer et al., 2017) that outputs a unique weight distribution for HP experts, depending on the current input feature, thereby exhibiting selectivity.

The adaptation process now is much more dynamic and flexible with intricate interaction between the adapters. With strong empirical evidence, we find our design enables weight updates that have a much more higher rank than LoRA. Furthermore, Lily enables a more comprehensive information access by allowing adapters at each layer to access information from other layers, promoting an interconnected and dynamic learning process, where the adapters can collaborate, share learned knowledge and model dependencies across layers. Overall, our contributions include:

- We propose Lily, a novel PEFT framework that incorporates cross-layer connections of the projection matrices, breaking the restriction of low-rank weight updates in LoRA.

- Lily utilizes routers to selectively connect an LP with multiple HP experts, enabling comprehensive information access and therefore expressive adaptation.

- Extensive experiments are conducted across various modalities, architectures, and model sizes, highlighting Lily's great performance and efficiency in diverse scenarios.

## 2 RELATED WORK

**Parameter Efficient Fine-Tuning** Typical usage of foundation models includes pre-training on large datasets and fine-tuning on various downstream tasks. Parameter-efficient fine-tuning (PEFT) thus emerges as a promising field, aiming to fine-tuning the model efficiently with minimal parameters while maintaining performance and preserving previously learned knowledge, addressing drawbacks posed by conventional fine-tuning techniques like full fine-tuning or linear probing. Current PEFT research can be mainly categorized into two types: 1) adapter-based methods (Hu et al., 2021), (Chen et al., 2022), (Pfeiffer et al., 2020b), (Jie & Deng, 2023) (Houlsby et al., 2019b) and 2) prompt-based methods (Tu et al., 2023b) (Tu et al., 2023a). Adapter-based methods introduce lightweight adapters into the Multi-Head Self-Attention (MHSA) or the Feed-Forward Network (FFN) blocks within the Transformer architecture. On the other hand, prompt-based methods append trainable tokens as prompts to the input sequence fed to certain parts of the model.

Among these various PEFT techniques, low-rank adaptation (LoRA (Hu et al., 2021)) stands out as one of the most well-known methods. LoRA introduces a pair of projection matrices $A$ and $B$ per adaptation target $W$. The low-dimension projector (LP) $A$ projects input $x$ to low-dimension space, and the high-dimension projector (HP) $B$ restores it to its original dimension. Multiplying these projection matrices approximates the weight update $\Delta W$ in FFT. Recent work (Hao et al., 2024) has shown that LoRA adapters are essentially performing random projection to the gradient using a fixed matrix. This restricts the learned weight update to low-rank subspace and thus imitating the model performance. Meanwhile, $A$ and $B$ are tightly coupled, therefore the adaptation process only has information access from current layer, without an understanding of information from other layers, which could be beneficial to modeling dependencies across various layers.

**Mixture of Experts** Mixture of Experts (MoE) is an active research area that has garnered significant attention, especially in the field of large language models (LLMs). Conditional computation, where different parts of the network are activated on a per-example basis, has been proposed to enhance model capability without increasing computation (Davis & Arel, 2013) (Bengio et al., 2013) (Eigen et al., 2013) (Almahairi et al., 2016). The sparsely-gated MoE layer is introduced to implement this idea, consisting of numerous sub-networks (Shazeer et al., 2017). A trainable gating network, known as a "router", determines the combination of experts for each example. There are already PEFT methods like MoLORA (Zadouri et al., 2023) and MOLA (Gao et al., 2024a) which apply the MoE design concept to PEFT. However, these methods simply treat the adapters combined in LoRA as a single expert. A concurrent research Wu et al. (2024), utilizes LP and HP sub-spaces as the experts but fails to overcome the limitation discussed in previous section. Another concurrent work, HydraLoRA Tian et al. (2024) also explores an asymmetric design for LoRA. A fundamental difference from our work is that we consider the interaction across layers from the model and deploy an model-wide asymmetric design to allow cross-layer connection.

## 3 METHODOLOGY

### 3.1 DOWNWARD PROJECTION AND SELECTIVE WEIGHT ALLOCATION

The process is illustrated in the right half of Fig. 1. Initially, we use an LP to project the input $x \in \mathbb{R}^{N \times C_{in}}$ into its low-dimensional representation $x' \in \mathbb{R}^{N \times d}$ where $N$ is the sequence length:

$$x' = xP_L \tag{1}$$

The number of LPs can be flexibly set, as discussed in A. Inspired by the Mixture of Experts (MoE) paradigm, we employ a router $R \in \mathbb{R}^{N_e \times d}$ to selectively assign weights to all HP experts based on their relationship to the current layer's features ($x'$). The weight set $S$ is obtained as:

$$S = softmax(\sum_{i=1}^{N}(x'R^T)_i) \tag{2}$$

The router selectively combines experts based on the current layer's features, enabling smart information integration. For shallower inputs, the router increases attention for experts specializing in shallow-layer knowledge, while deeper inputs favor experts learning deep-layer knowledge.

### 3.2 WEIGHTED COMBINATION OF EXPERTS AND UPWARD PROJECTION

Once we obtain the low-dimensional input $x'$, we combine information from all layers using the model-wide shared global HP module. One intuitive approach is to feed $x'$ into each HP expert and combine their outputs to obtain the extra knowledge $x_\Delta \in \mathbb{R}^{N \times C_{out}}$. However, to address efficiency concerns discussed in Appendix A.2, we propose an alternative implementation that is mathematically equivalent and significantly reduces computational burden, described as:

$$x_\Delta = x'(\sum_{i=1}^{N_e} S_i \cdot P_H^i) \tag{3}$$

where $S \in \mathbb{R}^{N_e}$ is the set of weight scores for HP experts, obtained through selective weight allocation. Since each $S_i$ is a scalar value, the calculation in Eq. 3 is mathematically equivalent to

Table 1: Commonsense reasoning results for Falcon-Mamba-7B across eight tasks. Bold represents the highest performance for each dataset utilizing PEFT methods.

| Model | PEFT | BoolQ | PIQA | SIQA | HellaSwag | WinoGrande | ARC-e | ARC-c | OBQA | Avg. |
|---|---|---|---|---|---|---|---|---|---|---|
| ChatGPT | - | 73.1 | 85.4 | 68.5 | 78.5 | 66.1 | 89.8 | 79.9 | 74.8 | 77.0 |
| | LoRA | 6.5 | 30.5 | 40.6 | **14.9** | 56.4 | 42.2 | 31.8 | 38.4 | 32.7 |
| Falcon-Mamba-7B | Lily ($\Delta$ + in) | 44.9 | **66.8** | 65.0 | 10.5 | 57.1 | 78.7 | 64.6 | **68.2** | 57.0 |
| | Lily (in) | **60.2** | 61.0 | **67.3** | 12.9 | **61.5** | **80.0** | 67.5 | 65.8 | **59.5** |

Table 2: Commonsense reasoning results for LLaMA3-8B across eight tasks. [†] represents results taken from Liu et al. (2024) and (Wang et al., 2024). Bold denotes the highest performance scores for each dataset among different PEFT methods.

| Model | PEFT | BoolQ | PIQA | SIQA | HellaSwag | WinoGrande | ARC-e | ARC-c | OBQA | Avg. |
|---|---|---|---|---|---|---|---|---|---|---|
| ChatGPT | - | 73.1 | 85.4 | 68.5 | 78.5 | 66.1 | 89.8 | 79.9 | 74.8 | 77.0 |
| | LoRA[†] | 70.8 | 85.2 | **79.9** | 91.7 | 84.3 | 84.2 | 71.2 | 79.0 | 80.8 |
| LLaMA3-8B | PiSSA[†] | 67.1 | 81.1 | 77.2 | 83.6 | 78.9 | 77.7 | 63.2 | 74.6 | 75.4 |
| | MiLoRA[†] | 68.8 | **86.7** | 77.2 | **92.9** | **85.6** | 86.8 | 75.5 | 81.8 | 81.9 |
| | Lily | **72.9** | 85.6 | 77.8 | 92.7 | 83.3 | **89.7** | **77.6** | **82.8** | **82.8** |

the intuitive implementation, but with significantly improved efficiency. Therefore, the whole computation flow, with input $x \in \mathbb{R}^{N \times C_{in}}$ and output $y \in \mathbb{R}^{N \times C_{out}}$, for an adaptation target module is:

$$y = xW_0 + s \cdot x_\Delta \tag{4}$$

$$= xW_0 + s \cdot xP_L(\sum_{i=1}^{N_e}(softmax(\sum_{j=1}^{N}(xP_L R^T)_j))_i \cdot P_H^i) \tag{5}$$

where $s$ is a scaling factor. By selectively allocating weights and combining HP experts, Lily enables access to all levels of information during adaptation. Each layer's target adaptation modules could consider the status and knowledge from all other layers, resulting in a more expressive and comprehensive adaptation. Meanwhile, thanks to its inter-connectivity and selectivity, Lily break the low-rank update constraint of LoRA and enable high-rank updates, as discussed in preliminaries.

## 4 EXPERIMENTS

We validate the effectiveness of Lily across different domains, model sizes (from ViT to LLM), and architectures (Transformers, Mamba), demonstrating its general strong adaptation capability. Concurrently, we conduct a comprehensive analysis of Lily's intrinsic mechanisms, providing a thorough understanding of Lily. All experiments are conducted on a single RTX 4090 GPU. Additionally, multiple analysis are provided in Appendix C, D, E, F, G, H, I and J.

### 4.1 COMMON SENSE REASONING

**Implementation** We evaluate Lily on commonsense reasoning with LLMs. Regarding the implementation, we utilize LLaMA3-8B (AI@Meta, 2024) and Falcon-Mamba-7B (Zuo et al., 2024) as backbones. LLaMA3 is a near-SOTA open-source large language model, while Falcon-Mamba is the latest and only open-source large language model based on the Mamba architecture. Using these models allows us to validate the effectiveness of Lily for fine-tuning LLMs and whether this effectiveness can be transferred to architectures beyond Transformers (Mamba, in this case). We fine-tune these models on Commonsense170K (Hu et al., 2023) and evaluate the adaptation results on eight multiple-choice problem tasks, including BoolQ (Clark et al., 2019), PIQA (Bisk et al., 2020), SIQA (Sap et al., 2019), HellaSwag (Zellers et al., 2019), WinoGrande (Sakaguchi et al., 2021), ARC-e, ARC-c (Clark et al., 2018), and OBQA (Mihaylov et al., 2018). The compared methods are LoRA for Falcon-Mamba and LoRA (Hu et al., 2021), PiSSA (Meng et al., 2024), and MiLoRA (Wang et al., 2024) for LLaMA3. We only compare LoRA for Falcon-Mamba because tailored PEFT methods for Mamba-based LLMs have not yet been proposed, which is beyond the scope of this paper. Detailed hyper-parameter settings and datsets information are reported in Appendix B.1.1 and Appendix B.2.1.

Table 3: Various fine-tuning methods applied to RoBERTa Base and RoBERTa Large are evaluated on 6 datasets from the GLUE benchmark. We present the Matthew's correlation coefficient (MCC) for CoLA, Pearson correlation coefficient (PCC) for STS-B, and accuracy (Acc.) for the remaining tasks. The highest performance for each dataset is highlighted in **bold**, with all metrics favoring higher values across the 6 datasets.

| Model & Method | # Trainable Parameters | SST-2 (Acc.) | MRPC (Acc.) | CoLA (MCC) | QNLI (Acc.) | RTE (Acc.) | STS-B (PCC) | Avg. |
|---|---|---|---|---|---|---|---|---|
| $RoB_{base}$(FFT) | 125M | 94.8 | 90.2 | 63.6 | 92.8 | 78.7 | 91.2 | 85.2 |
| $RoB_{base}$(BitFit) | 0.1M | 93.7 | **92.7** | 62 | 91.8 | 81.5 | 90.8 | 85.4 |
| $RoB_{base}$(Adpt$^D$) | 0.3M | 94.2 | 88.5 | 60.8 | 93.1 | 71.5 | 89.7 | 83.0 |
| $RoB_{base}$(Adpt$^D$) | 0.9M | 94.7 | 88.4 | 62.6 | 93.0 | 75.9 | 90.3 | 84.2 |
| $RoB_{base}$(LoRA) | 0.3M | **95.1** | 89.7 | 63.4 | **93.3** | 78.4 | **91.5** | 85.2 |
| $RoB_{base}$(AdaLoRA) | 0.3M | 94.5 | 88.7 | 62.0 | 93.1 | 81.0 | 90.5 | 85.0 |
| $RoB_{base}$(DyLoRA) | 0.3M | 94.3 | 89.5 | 61.1 | 92.2 | 78.7 | 91.1 | 84.5 |
| $RoB_{base}$(**Lily**) | 0.3M | 95.0 | 90.2 | **66.0** | 92.5 | **81.6** | 90.8 | **86.0** |
| $RoB_{large}$(FF) | 356M | **96.4** | **90.9** | 68 | 94.7 | 86.6 | **92.4** | 88.2 |
| $RoB_{large}$(Adpt$^H$) | 0.8M | 96.3 | 87.7 | 66.3 | 94.7 | 72.9 | 91.5 | 84.9 |
| $RoB_{large}$(LoRA) | 0.8M | 96.2 | 90.2 | 68.2 | **94.8** | 85.2 | 92.3 | 87.8 |
| $RoB_{large}$(Lily) | 0.5M | 95.6 | **90.9** | **68.4** | **94.8** | **88.4** | 91.9 | **88.4** |

**Results** We report the accuracy in the Table 2 and Table 1. Based on the results, it is evident that Lily performs the best out of the compared PEFT methods. Lily surpasses LoRA by a significant margin on Falcon-Mamba, and on LLaMA3, it outperforms LoRA and MiLoRA. This indicates Lily's superior adaptation capability and parameter efficiency dealing with commonsense reasoning tasks. Additionally, while the performance on Falcon-Mamba is notably lower than the baseline and LLaMA3, we believe this is due to the model's limitations rather than Lily's, as Lily still significantly outperforms LoRA on Falcon-Mamba and demonstrates great performance on LLaMA3. This sheds light on the current state of Mamba-based LLMs, showing that they generally have inferior performance compared to Transformer-based LLMs like ChatGPT and LLaMA on many tasks.

## 4.2 NATURAL LANGUAGE UNDERSTANDING

**Implementation** We evaluate Lily on natural language understanding (NLU) tasks. For implementation, we use RoBERTa Base (Liu et al., 2019) and RoBERTa Large as the backbones and fine-tune them on tasks from GLUE benchmark (General Language Understanding Evaluation (Wang et al., 2018)), consisting of multiple NLU tasks including single-sentence classification tasks, similarity and paraphrase tasks and natural language inference tasks. We compare Lily against several competitive PEFT methods, including BitFit (Zaken et al., 2021), Adapter-Tuning (Rücklé et al., 2020) (Houlsby et al., 2019a) (Lin et al., 2020) (Pfeiffer et al., 2020a), LoRA (Hu et al., 2021), DyLoRA (Valipour et al., 2022) and AdaLoRA (Zhang et al., 2023). Additionally, we utilize full fine-tuning (FFT) as the baseline. Specific hyper-parameters and datasets information are provide in Appendix B.1.2 and B.2.2.

**Results** The results are shown in Table. 3, from which we can clearly observe that Lily surpass all of the compared PEFT methods by a significant margin, demonstrating its capability of tackling NLU tasks. Among the 6 given tasks, Lily surpasses FFT in 4 of them using RoBERTa-Base and RoBERTa-Large, showcasing its strong approximation ability with high-level parameter-efficiency.

## 4.3 SUBJECT-DRIVEN IMAGE GENERATION

**Implementation** We conduct experiments on fine-tuning text-to-image diffusion models for the subject-driven generation task (Ruiz et al., 2023). For backbone, we use SDXL and we fine-tune it using LoRA and Lily. We first fine-tune the model with images associated with text prompts (e.g., A photo of a [v] duck toy), in which a unique identifier is provided. After that, text prompts containing the identifier could be used to generate customized images.

**Results** The results are presented in Fig. 2 following the format in Gao et al. (2024b) and Wu et al. (2024), from which we can observe that images generated by Lily generally align better with the

Figure 2: Results of subject-driven generation. Lily's results align better with prompts, featuring more accurate color, environment, and shape.

Table 4: Full results of Lily on ViT-B pre-trained on ImageNet-21K for the VTAB-1K benchmark, with averages computed based on group-wise results. **Bold** indicates the best performance.

| | Params(M) | Average | Natural | | | | | | | Specialized | | | | Structured | | | | | | | |
| | | | Cifar100 | Caltech101 | DTD | Flowers102 | Pets | SVHN | Sun397 | Camelyon | EuroSAT | Resisc45 | Retinopathy | Clevr-Count | Clevr-Dist | DMLab | KITTI-Dist | dSpr-Loc | dSpr-Ori | sNORB-Azim | sNORB-Ele |
|---|---|---|---|---|---|---|---|---|---|---|---|---|---|---|---|---|---|---|---|---|---|
| *Conventional Fine-Tuning* | | | | | | | | | | | | | | | | | | | | | |
| FFT | 86 | 68.9 | 68.9 | 87.7 | 64.3 | 97.2 | 86.9 | 87.4 | 38.8 | 79.7 | 95.7 | 84.2 | 73.9 | 56.3 | 58.6 | 41.7 | 65.5 | 57.5 | 46.7 | 25.7 | 29.1 |
| LP | 0 | 57.6 | 64.4 | 85.0 | 63.2 | 97.0 | 86.3 | 36.6 | 51.0 | 78.5 | 87.5 | 68.5 | 74.0 | 34.3 | 30.6 | 33.2 | 55.4 | 12.5 | 20.0 | 9.6 | 19.2 |
| *PEFT methods* | | | | | | | | | | | | | | | | | | | | | |
| AdaptFormer | 0.588 | 76.8 | 74.0 | 92.2 | 71.7 | **99.3** | **91.7** | 88.9 | 56.4 | 87.2 | 95.1 | **85.7** | **75.9** | **84.2** | 62.2 | 53.0 | 81.0 | 87.1 | 53.6 | 35.3 | 42.3 |
| Bi-LoRA | 1.180 | 76.7 | 72.1 | 91.7 | 71.2 | 99.1 | 91.4 | **90.2** | 55.8 | 87.0 | **95.4** | 85.5 | 75.5 | 83.1 | 64.1 | 52.2 | 81.3 | 86.4 | 53.5 | 36.7 | 44.4 |
| LoRA | 1.180 | 76.4 | 72.5 | 91.5 | 71.9 | 99.1 | 91.4 | 89.6 | 56.0 | 87.6 | 95.3 | 84.0 | 75.0 | 83.6 | 64.3 | 51.6 | 80.9 | 86.0 | 51.8 | **36.8** | 42.3 |
| FourierFT | 0.936 | 72.7 | 69.1 | 88.8 | 71.9 | 99.0 | 91.0 | 79.0 | 55.6 | 84.9 | 93.0 | 83.2 | 74.9 | 70.7 | 61.1 | 45.2 | 74.8 | 78.0 | 53.0 | 24.8 | 30.8 |
| MoRA | 1.058 | 75.4 | 72.1 | 90.0 | 71.7 | 99.2 | 91.1 | 90.1 | 56.0 | 87.1 | 94.8 | 85.1 | 75.4 | 76.7 | 62.3 | 49.7 | 78.3 | 83.1 | 53.0 | 34.5 | 34.5 |
| Lily | 0.318 | **77.3** | **73.9** | **93.0** | **72.9** | 99.3 | 91.6 | 89.0 | **56.6** | **87.9** | 95.2 | 84.9 | 75.7 | 83.9 | **65.4** | **53.4** | **81.6** | **88.2** | **54.5** | 37.0 | **45.4** |

text prompts. For instance, when asked to generate a duck toy floating on top of water, Lily's image accurately depicts the designated environment, whereas LoRA's does not. Additionally, when asked to generate a wolf plushie in snow, Lily precisely depicts the snow around the wolf, while LoRA fails to do so. These observations demonstrate Lily's excellent ability in the domain of text-to-image generation with more expressive adaptation. More generated results are in Appendix I.

## 4.4 VISUAL ADAPTATION BENCHMARK

**Implementation** We assess Lily on the Visual Task Adaptation Benchmark (VTAB-1K Zhai et al. (2019)), a suite of 19 visual tasks spanning diverse domains and semantics, to test its general visual adaptation capability. Tasks are categorized into Natural, Specialized, and Structured, all formulated as classification problems for consistent model evaluation. We conduct two sets of experiments: one focusing on the adaptation effectiveness on Vision Transformer (ViT (Dosovitskiy et al., 2020)) and the other on Vision Mamba (Vim (Zhu et al., 2024)), demonstrating Lily's architecture-agnostic capabilities. For ViT, we use ViT-B pre-trained on ImageNet-21K (Deng et al., 2009), and for Vim, Vim-s pre-trained on ImageNet-1K. To fairly compare ViT and Vim architectures, we implement LoRA (Hu et al., 2021) and AdaptFormer (Chen et al., 2022) on ViT-B pre-trained on ImageNet-1K. In ViT experiments, we compare Lily with LoRA, AdaptFormer, FourierFT (Gao et al., 2024b),

Table 5: Full results of Lily on Vim-S pre-trained on ImageNet-1K for the VTAB-1K benchmark, with averages calculated within each group. * denotes linear probing results from Tu et al. (2023b). For fair comparison, we also use ViT-B pre-trained on ImageNet-1K. **Bold** indicates best performance among Vim-based PEFT methods.

| | Params(M) | Average | Cifar100 | Caltech101 | DTD | Flowers102 | Pets | SVHN | Sun397 | Camelyon | EuroSAT | Resisc45 | Retinopathy | Clevr-Count | Clevr-Dist | DMLab | KITTI-Dist | dSpr-Loc | dSpr-Ori | sNORB-Azim | sNORB-Ele |
|---|---|---|---|---|---|---|---|---|---|---|---|---|---|---|---|---|---|---|---|---|---|
| | | | | | Natural | | | | | Specialized | | | | Structured | | | | | | | |
| *Conventional Fine-Tuning* | | | | | | | | | | | | | | | | | | | | | |
| FFT-Vim | 26 | 70.1 | 47.7 | 89.4 | 64.2 | 89.0 | 87.7 | 90.6 | 35.1 | 84.5 | 93.9 | 81.0 | 74.5 | 67.5 | 52.9 | 47.3 | 78.9 | 75.3 | 53.9 | 33.3 | 29.4 |
| FFT-ViT | 86 | 69.9 | 49.4 | 89.3 | 65.5 | 91.7 | 89.1 | 91.4 | 33.5 | 85.9 | 93.6 | 85.4 | 74.3 | 54.7 | 55.2 | 48.7 | 79.7 | 68.2 | 49.7 | 31.5 | 27.7 |
| LP-Vim | 0 | 55.3 | 40.9 | 83.3 | 57.3 | 66.3 | 86.3 | 38.4 | 34.6 | 79.0 | 87.6 | 65.0 | 73.6 | 36.3 | 35.1 | 33.3 | 64.8 | 23.0 | 21.6 | 15.1 | 21.7 |
| LP-ViT | 0 | 66.4 | 50.6 | 85.6 | 61.4 | 79.5 | 86.5 | 40.8 | 38.0 | 79.7 | 91.5 | 71.7 | 65.5 | 41.4 | 34.4 | 34.1 | 55.4 | 18.1 | 26.4 | 16.5 | 24.8 |
| *PEFT on ViT* | | | | | | | | | | | | | | | | | | | | | |
| AdaptFormer | 0.147 | 72.4 | 56.2 | 89.6 | 67.2 | 91.2 | 91.1 | 85.9 | 42.1 | 85.4 | 94.6 | 84.0 | 74.3 | 75.8 | 58.6 | 48.6 | 79.6 | 81.6 | 53.7 | 29.6 | 35.2 |
| LoRA | 0.295 | 72.5 | 56.4 | 89.0 | 66.9 | 91.2 | 90.4 | 86.9 | 41.5 | 85.4 | 95.1 | 84.1 | 75.2 | 75.8 | 61.7 | 47.7 | 80.5 | 80.4 | 52.0 | 29.4 | 35.7 |
| *PEFT on Vim* | | | | | | | | | | | | | | | | | | | | | |
| LoRA | 0.054 | 70.1 | 57.5 | 87.7 | 64.4 | 86.0 | 90.0 | 85.7 | 39.8 | 82.2 | 93.8 | 79.6 | 72.5 | 78.6 | 56.5 | 42.0 | 80.5 | 71.8 | 51.0 | 28.4 | 32.6 |
| Lily-S | 0.074 | 71.4 | **58.2** | 88.5 | 65.6 | 87.1 | **90.7** | 87.5 | 40.4 | 83.3 | 94.1 | 79.7 | 73.8 | 81.2 | 57.3 | 44.1 | 80.9 | 79.3 | 54.1 | 30.0 | 33.7 |
| Lily-L | 0.196 | **72.3** | 57.8 | **89.4** | **66.2** | 87.8 | 90.5 | **88.1** | **40.5** | **84.1** | **94.3** | **81.3** | **75.1** | **81.6** | **57.8** | **46.5** | **81.0** | **82.9** | **55.2** | **32.1** | **34.8** |

and MoRA (Jiang et al., 2024); in Vim experiments, we focus on contrasting architectures difference, therefore only using LoRA as the baseline. All experiments include FFT and linear probing as baselines. For Vim, we implement two versions: Lily-S (Small) and Lily-L (Large) of Lily, with different hyperparameter settings to either reduce the parameter count (Lily-S) or maximize performance (Lily-L). For Lily on ViT, the reported results are obtained from adapting both the self attention and the MLP module in Transformer. For the performance w.r.t the fine-tuned module, we conduct additional experiments in Appendix D. Detailed experimental settings and datasets information are provide in Appendix B.1.3 and B.2.3.

**Results** Results are shown in Table 4 and Table 5. For ViT, Lily significantly outperforms all compared PEFT methods with improved parameter efficiency. For Vim, results on ViT generally surpass those on Vim. For instance, LoRA on ViT performs better than LoRA on Vim. We argue that this is due to differences in architecture designs and general model sizes. However, Lily's strong adaptation performance allows it to match or exceed PEFT methods on ViT and significantly outperform LoRA on Vim (Lily-S and Lily-L surpass LoRA by a significant margin). This demonstrates Lily's architecture-agnostic capability, highlighting its potential across various model architectures. In general, Lily has achieved great visual adaptation capability with an advantage of being architecture-agnostic and enjoying excellent parameter-efficiency.

## 4.5 Understanding Lily

### 4.5.1 Does Lily Has High-Rank Weight Updates?

We state that Lily achieves weight updates that with higher rank than LoRA. To validate our claim, we provide an empirical analysis as shown in Fig. 3. Specifically, we run 4 tasks from the NLU experiment and test the rank of the weight updates for $W_q$ in the first three layers. We uses small number of LPs and HPs (2 or 3) to demonstrate the efficiency and to match the parameter count. Specific hyperparameter settings can be found in Appendix B.1.2.

From the results, we can observe that the rank of weight updates from Lily generally is notably larger than LoRA when using a similar amount of parameters. Meanwhile, weight updates from Lily still have higher rank compared to LoRA even when using 16.7% of the parameters of LoRA. This empirical analysis essentially validate our claim that Lily achieves high-rank updates with the same parameter budget. We credit it to the model-wide sharing mechanism and the cross-layer asymmetric design, which facilitates dynamic and expressive adaptation.

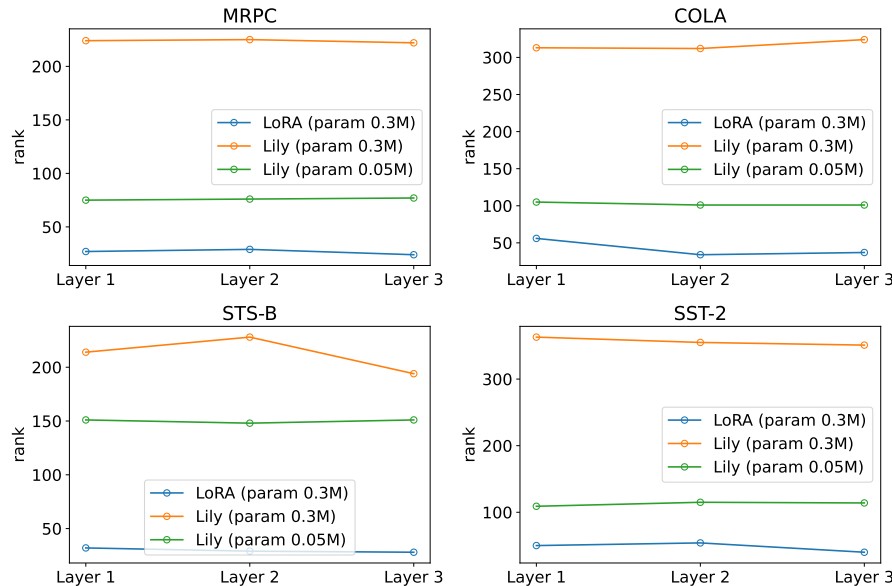

Figure 3: Actual rank of the weight updates. We run 20 epochs for COLA, MRPC and STS-B and 3 epochs for SST-2. It can be easily observed that weight updates from Lily have notably higher rank than LoRA.

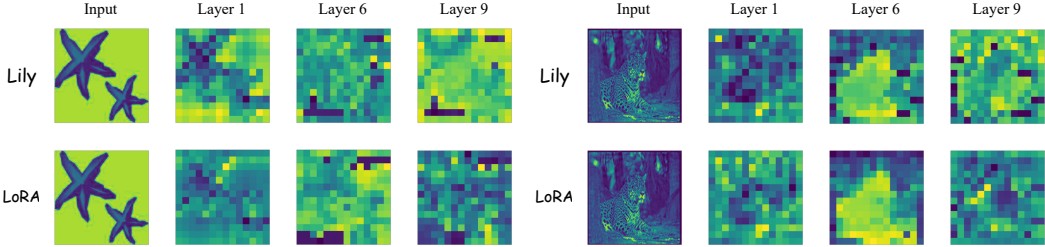

Figure 4: Attention maps of Lily and LoRA. The input images for the example here are taken from Caltech101 datasets from VTAB-1K benchmark. It can be observed that features from a certain layer have more similarity to those in other layers in Lily than in LoRA.

### 4.5.2  FROM A FEATURE MERGING PERSPECTIVE

Apart from having higher-rank weight updates than LoRA, Lily also enables comprehensive information access across layers. Lily enables access to information or features from all other layers when adapting a target module at a specific layer thanks to the inter-connectivity of the adapters. We aim to understand how Lily achieves this comprehensive information access from the perspective of visual tasks as shown in Fig. 4. We can observe that, in Lily, the distinctness of the attention maps between layers is not as pronounced as in LoRA. This validates Lily's ability to enable all-level information access, since adaptation at each layer takes into account features from other layers. Additionally, we specifically visualize the actual feature differences between different layers in Fig. 5. We observe that Lily has more points with low feature differences (blue color) than LoRA, indicating that the distinctness of features between layers in Lily is generally lower than in LoRA. This further demonstrates Lily's ability to enable comprehensive information access. Although we enable all-level information access, what prevents the features from becoming completely identical is the selectivity introduced by Lily, which we specify in the following section.

### 4.5.3  WHAT'S THE INFLUENCE OF ATTENTION GRANULARITY?

The number of experts in the model-wide HP module can be freely set, and the number of LPs can also be flexibly set by sharing across the same level of layers introduced in Appendix A.1. Therefore,

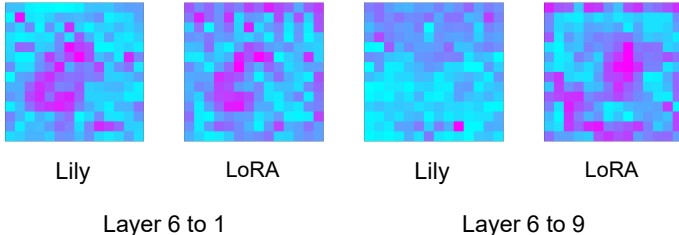

Lily LoRA Lily LoRA

Layer 6 to 1 Layer 6 to 9

Figure 5: Feature difference measured in absolute distance for each element. We compare Lily and LoRA in terms of the difference between features from different layers. In this example image taken from Caltech101, we visualize the feature difference between layers 6 and 1, as well as between layers 6 and 9.

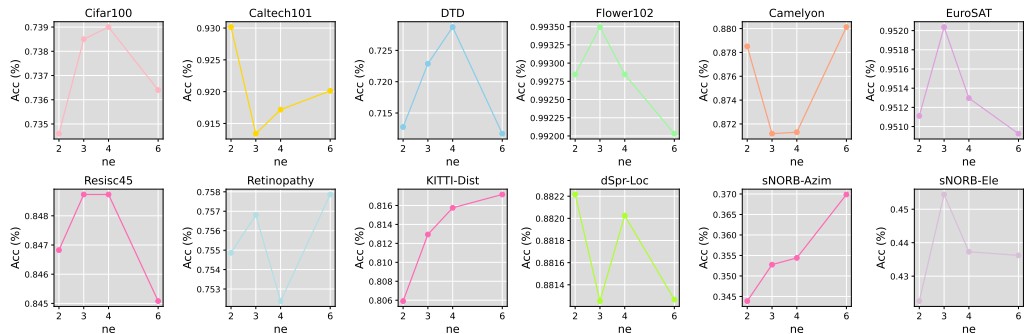

Figure 6: Impact of attention granularity (i.e., the choice of how many LPs and HPs) on the performance. We choose 12 out of 19 tasks from VTAB-1K for a comprehensive understanding.

we analyze the impact of these choices on performance. We denote the number of LP experts and HP experts as $ne\_1$ and $ne\_2$, respectively. For simplicity, we make them identical in the experiments, denoted as **ne**. We refer to the number of layers each expert attends to as attention granularity. As the value of *ne* increases, the attention granularity becomes finer. As shown in Fig. 6, the results from the VTAB-1K benchmark indicate different patterns. For instance, on the DTD dataset, the best performance is achieved when *ne* is 4, while on sNORB-Azim, performance increases with the increase in *ne*. Increasing *ne* leads to more parameters and finer attention granularity. However, finer attention granularity does not necessarily lead to better overall performance. For example, on Resisc45, DTD, Cifar100, sNORB-Ele, dsPr-LoC, Flowers102, and EuroSAT, the negative impact of increasingly finer attention granularity eventually outweighs the benefits of increased parameters, leading to a decrease in overall performance. In other tasks, different patterns may occur because the positive effect of attention granularity on performance is consistently strong, or its negative effect is not enough to offset the benefits of increased parameters, resulting in a generally increasing performance with *ne*. This phenomenon provides an important insight: for most tasks, simply increasing parameters may not lead to better performance. Instead, only when attention granularity and the number of parameters reach a good tradeoff can we achieve the best performance.

### 4.5.4 DOES LILY EXHIBIT SELECTIVITY?

Lily uses routers to assign varying weights to different HP experts, thereby achieving selective information combination. We illustrate this selectivity in Fig. 8. We use a setup with three HP experts and select three layer levels (1, 13, 22) to calculate the total weight assigned to each expert. The results reveal a clear selectivity: for different layers, the router assigns significantly different weights to different HP experts. For instance, on Cifar100, the middle layer is predominantly dominated by HP 2, whereas the deep layer is primarily dominated by HP 1 and HP 2. In contrast, on Retinopathy, both the middle and deep layers are dominated by HP 3. This selectivity ensures that, even when different layers share information, the inherent differences between layers are still taken into account, making the adaptation more flexible and comprehensive.

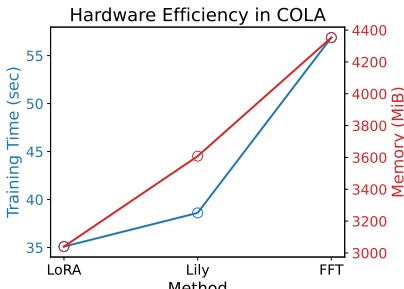 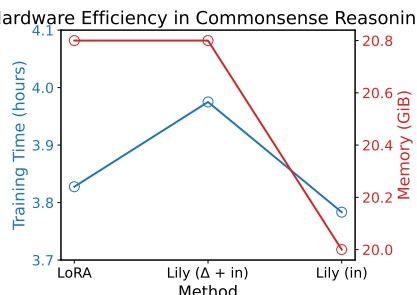

Figure 7: Hardware efficiency of Lily compared to LoRA. We run 10 epochs for COLA. We report the training time and memory consumption. It can be observed that Lily generally performs on par with LoRA in terms of hardware efficiency.

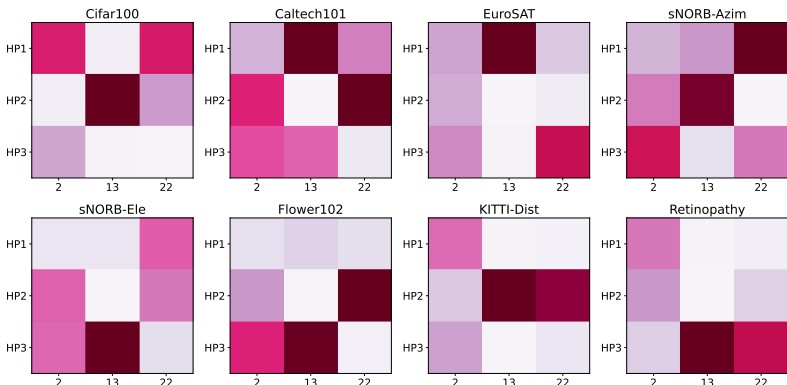

Figure 8: Visualization of accumulated assigned weight for HP experts by a router across various layers. Example here uses layer of index 2, 13 and 22 to represent shallow, middle and deep layers.

### 4.5.5 WHAT'S THE HARDWARE EFFICIENCY OF LILY?

The dynamic of Lily obviously introduces complexity onto the design of LoRA. In this section, we analyses how does this affect the hardware efficiency of Lily compared to LoRA. We use the COLA task from the NLU experiments using RoBERTa-Base and run 10 epochs. Additionally, we also report the runtime and GPU memory consumption in the Falcon-Mamba experiment.

The results are shown in Fig. 7, from which we can observe that the hardware efficiency of Lily is comparable to LoRA. Specifically, Lily slightly under-perform LoRA in the NLU experiment but performs on par with LoRA in the LLM experiment. In generally, the introduced complexity of Lily does not prevent it from being a effective PEFT method that is hardware-friendly.

## 5 CONCLUSION

In this paper, we propose low-rank interconnected adaptation (Lily), a novel framework for efficient fine-tuning via inter-connectivity of adapters. Lily enables each layer to access information from others during adaptation through a hierarchical structure. Additionally, it successfully overcome the low-rank update limitation of LoRA, enabling high-rank update and therefore better adaptation capability. Our approach consistently improves performance across various modalities, model sizes, and architectures, surpassing existing methods with enhanced efficiency. In summary, Lily's versatility and efficiency make it a promising approach for a wide range of applications.

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

APPENDIX

# A   MORE DISCUSSION ABOUT LILY

## A.1   MODEL STRUCTURE AND DESIGN INTUITION OF LILY

Within the overall framework of Lily, we delve into specific implementation details and model design insights. First, we have established the relationship between LPs and HP: LP is confined to specific levels of layers, capturing features that enable the router to selectively assign weights to the HP experts. In contrast, HP is a model-wide module comprising multiple experts, each of which contains information from a particular level of layers. We highlight several key aspects which are not heavily discussed in the methodology section:

### A.1.1   NUMBER OF LPS

Since LP is limited to specific layers, the simplest approach would be to place an LP at each layer of the module to be adapted (e.g., the query transformation in MHSA). However, this setup may not be necessary: the importance of each layer varies, and many layers have significantly lower importance than others (Zhang et al., 2023). To achieve greater parameter efficiency, we can set up fewer LPs, with each LP focusing on a level of layers rather than a single layer. For example, an LP can focus on shallow layers (e.g., layers 0, 1, 2, etc.) or deep layers. To enable a single LP to handle multiple layers, we can share an LP across multiple layers. By doing so, we eliminate the redundancy of having an LP at each layer, reduce the number of parameters, and increase efficiency. **This is exactly the strategy adopted by most of the experiments.**

### A.1.2   NUMBER OF HP EXPERTS

Regarding HP, the number of experts can be arbitrarily set, enabling more flexible configurations. In our experiments, for the sake of simplicity, we set the number of HP experts equal to the number of LPs, thereby equating the granularity of LP and HP.

### A.1.3   ROUTERS SETUP

There are also different settings that can be employed for the router. First, we can bind the router to HP, resulting in only one router per model. However, since the number of parameters in the router is relatively small, having only one router per model may not lead to significant selectivity. Therefore, we can also bind the router to LP, configuring a separate router for each LP. Most of our experiments use the latter setup, but in the vision experiments on Vim, we use the single-router and no-lp-sharing setup to verify its effectiveness. From the results, we can see that this setup also performs well. As future work, we can verify the effectiveness of using the latter setup on Vim, which may potentially lead to superior performance.

### A.1.4   HYPERPARAMETERS

We detail the hyperparameters used in Lily. Specifically, we use **Lily_r** to represent the hidden-dimension of the projectors: LPs and HPs. It serves the same function as r in LoRA. We use **Lily_s** to represent the scaling factor used by Lily. It is mostly searched within the range of $\{0.01, 0.1, 1.0, 10.0, 100.0\}$. We use **ne_1** to represent the number of LPs used in the model. Since the LPs can be shared as discussed in the previous section, $ne\_1$ does not need to equal the number of layers in the model. We use **ne_2** to represent the number of HP experts in the model-wide HP module. In our experiments, we set $ne\_1 = ne\_2$ to improve parameter-efficiency and simplicity.

### A.1.5   DESIGN INTUITION

Lily employs a hierarchical structure to enable updates with higher-ranks than LoRA. However, simply equally connecting all the HPs to the LPs can not achieve the best performance. From the perspective of feature and information utilization across layers, simply aggregating all HPs for an LP ignores the distinctness of the features from current layers. Meanwhile, it reduced the variability of the combinations of gradient projection matrices ($S_i$ and $C_{i,j}$ are constants now), making the rank of

the weight update higher than that of LoRA (since multiple distinct random matrices are used), but still not high enough for the best performance because of the lack of variability during combination. Therefore, we introduce selectivity into the inter-connectivity as well as discussed below, making the combination of HPs data-dependent so that each $S_i$ is unique across the time-steps, enabling updates with even higher ranks. The approach has similarity to Hao et al. (2024), where random matrix is constantly resampled to ensure updates with higher ranks. We further conduct an analysis in Appendix G.

## A.2 Efficient Implementation for Weighted Combination

One intuitive implementation of the weighted combination in Lily is make the inputs go through all the experts and then sum the results up, from which we could observe that it perform $N_e$ times of matrix multiplication, $N_e$ times of scalar multiplication and $N_e$ times of matrix addition. Therefore, despite its intuitive nature, the computational burden of this approach is quite formidable.

However, Eq. 3 which is adopted in Lily only utilize $N_e$ times of scalar multiplication, $N_e$ times of matrix addition and 1 time of matrix multiplication. This saves roughly $N_e$ times of matrix multiplication, which can be significant as the size of the model and number of adaptation targets increases. For a $x'$ size of $\mathbb{R}^{N \times d}$ and a $P_H \in \mathbb{R}^{d \times C}$, the floating-point operation (FLOPs) of these two implementation are:

$$\text{FLOPs} = \sum_{1}^{N_e}(2NdC) + \sum_{1}^{N_e}(dC) + \sum_{1}^{N_e}(NC)$$
$$= N_e \times (2NdC + dC + NC) \qquad \text{(Intuitive)}$$
$$\text{FLOPs} = 2\sum_{i=1}^{N_e}(dC) + 2NdC$$
$$= 2dC \times (N + N_e) \qquad \text{(Lily)}$$

(6)

from which we can easily observe that the approach adopted by Lily requires less computation and therefore provides more speed and efficiency during the fine-tuning process. Under the setting of $N = 1024, d = 16, C = 768, N_e = 4$, the FLOPs of the intuitive approach would be $0.104$ GFLOPs while in Lily it is merely $0.025$ GFLOPs, which could potentially lead to a 4X increase in speed.

## A.3 Actual implementation of Lily

We present the actual implementation of Lily in Fig. 9. For the example here, we choose the implementation from visual adaptation tasks (i.e., VTAB-1K benchmark). For LLM, the implementation is a bit more complicated because of modifications to the huggingface PEFT library (Mangrulkar et al., 2022), but the fundamental adaptation process is the same. Specifically, given an input, we first use the corresponding LP of the current layer to project it to a low-dimensional representation. After that, we use the low-dimensional representation to selectively assign weights for the HP experts. Once we obtain all the weights for the experts, we set out to combine these HP experts accordingly, as discussed in Appendix A.2. After the weighted combination, we use the obtained combined HP to project the low-dimensional representation to high-dimension, therefore acquiring the extra knowledge gained through adaptation.

# B Experimental settings

## B.1 Hyper-parameters

A detailed description of the hyper-parameters used in Lily is in Appendix A.1.

### B.1.1 Commonsense Reasoning

The hyper-parameters used in commonsense reasoning experiments for MiLoRA, PiSSA are provided in Table 7, 6. Settings for Lily and LoRA using Falcon-Mamba as the backbone are provided in Table. 9 and 8. It can be noticed that Lily achieves the best performance by merely adapting the

```python
class lily_adapter(nn.Module):
    """
    Implementation of a Lily adapter for an adaptable target. For symplicity, we assume that the
    ↪  number of hp expert is equal to the number of LPs.

    args:
        hidden_dim: hidden dimension
        ne: number of experts
        lp: low-dimensioanl projector
        hps: high-dimensioanl projector experts
        mlp: whether the adpatation target is located in MLP
    """
    def __init__(self, hidden_dim, ne, lp, hps, mlp=False):
        super().__init__()
        self.hps = hps
        self.ne = ne
        self.lp = lp
        self.router = nn.Linear(hidden_dim, ne, bias=False)
        if mlp:
            self.non_linear = nn.ReLU()
        else:
            self.non_linear = nn.Identity()
    def forward(self, x):
        hidden = self.non_linear(self.lp(x))
        router_logits = self.router(hidden) # [B, N, num_of_experts]
        router_probability = F.softmax(router_logits, dim=-1) # [B, N, ne]
        expert_probabilities = router_probability.mean(dim=(0, 1))
        combined_hp = torch.einsum("e,eio->io", expert_probabilities, self.hps)
        return torch.matmul(hidden, combined_hp)
```

Figure 9: Implementation of Lily in VTAB-1K benchmark.

multi-head self attention module (MHSA) in LLaMA3-8B, while other compared methods adapt all the modules including MLP. Meanwhile, Lily employs the least amount of parameters, showcasing its excellent adaptation at low parameter-budget scenarios.

Table 6: Hyperparameter configuration from the MiLoRA paper.

| MiLoRA hyperparameters | |
| --- | --- |
| Rank r | 32 |
| $\alpha$ of LoRA | 64 |
| $\alpha$ of PiSSA | 32 |
| Dropout | 0.05 |
| Optimizer | AdamW |
| LR | 3e-4 |
| LR Scheduler | Linear |
| Batch Size | 16 |
| Warmup Steps | 100 |
| Epochs | 3 |
| Placement | query, key, value, MLP up, MLP down |

### B.1.2 NATURAL LANGUAGE UNDERSTANDING

Specific hyper-parameter settings of Lily on GLUE benchmark are provided in Table 10. We fix the learning rate of both the backbone and the head as 5E-3 and tune the scaling factor Lily_s $\in \{0.01, 0.1, 1.0\}$ instead. For the rank $r$ we fix it to 32 and the seed to 0. The baseline results are taken from FourierFT Gao et al. (2024b).

### B.1.3 VISUAL ADAPTATION BENCHMARK

We provide the hyper-parameter for Lily on VTAB-1K benchmark in Table 11. Specifically, we fix the learning rate at 1E-3 with a weight decay of 1E-4. For ViT, we tune the scaling factor Lily_s $\in \{0.01, 0.1, 1.0, 10.0\}$ to maximize the performance, following Jie et al. (2023), Jie & Deng (2023).

Table 7: Hyperparameter configuration from the PiSSA paper.

| PiSSA hyperparameters | |
|---|---|
| $\alpha$ | Same as rank r |
| Dropout | 0.0 |
| Optimizer | AdamW |
| LR | 2e-5 |
| LR Scheduler | cosine |
| Batch Size | 128 |
| Warmup Ratio | 0.03 |
| Epochs | 1 |
| Placement | query, key, value, output, gate, MLP up, MLP down |

Table 8: Hyperparameter configuration for LoRA using Falcon-Mamba as backbone.

| LoRA hyperparameters | |
|---|---|
| Rank r | 2 |
| $\alpha$ | 16 |
| Dropout | 0.05 |
| Optimizer | AdamW |
| LR | 3e-4 |
| LR Scheduler | Linear |
| Batch Size | 16 |
| Epochs | 1 |
| Placement | input, delta |

For Vim, we fix Lily_s to 1.0. Additionally, we search for the hyper-parameters *ne_*1 and *ne_*2 within the range {2, 3, 4}, as these numbers can divide the number of layers in the ViT model (12 in ViT-B). For vim, we use the implementation discussed in section A.1, which does not share LPs across layers. Therefore, *ne_*1 in this setting is fixed to number of layers in Vim (22 in this case), while we search *ne_*2 in {3,6} and {5,6,17} separately for Lily-S and Lily-L. Note that *ne* is only set for input projection in Vim. For delta transformation, we only use a single HP expert to reduce the parameter cost. In the experiments of ViT, the rank $r$ is fixed at 16. Meanwhile in Vim's setting, we tune the ranks $r$ for the delta transformation module and the input projection module separately. We use $4, 4$ and $4, 8$ separately for Lily-S and Lily-L.

## B.2 DATASETS

### B.2.1 COMMONSENSE REASONING

We provide a short description of each datasets used in commonsense reasoning experiments in Table 12.

### B.2.2 NATURAL LANGUAGE UNDERSTANDING

We provide detailed information about datasets in the GLUE benchmark in Table 13.

### B.2.3 VISUAL ADAPTATION BENCHMARK

We provide detailed information about all the tasks from VTAB-1K benchmark in Table 14.

## C DOES SHARING LP RESULTS IN INFERIOR PERFORMANCE?

As mentioned earlier, we adopted a strategy of sharing the LP across most of our experiments, ensuring that the number of LP and HP experts is consistent. This approach offers two benefits:

Table 9: Best Hyperparameter configuration for Lily using Falcon-Mamba and LLaMA3 as backbones.

|  | **Falcon-Mamba** | **LLaMA3** |
|---|---|---|
| Rank r | 40 | 16 |
| $ne\_1$ | 4 | 4 |
| $ne\_2$ | 4 | 4 |
| Dropout | 0 | 0 |
| Optimizer | AdamW | AdamW |
| LR | 3e-4 | 3e-4 |
| LR Scheduler | Linear | Linear |
| Batch Size | 16 | 16 |
| Epochs | 1 | 3 |
| Placement | input | query, key, value |

Table 10: Hyperparameter of Lily on GLUE benchmark.

| Hyperparameter | STS-B | RTE | MRPC | CoLA | SST-2 | QNLI | MNLI | QQP |
|---|---|---|---|---|---|---|---|---|
| Optimizer | | | | AdamW | | | | |
| LR Schedule | | | | Linear | | | | |
| Learning Rate (Lily) | | | | 5e-3 | | | | |
| Learning Rate (Head) | | | | 5e-3 | | | | |
| Max Seq. Len | 512 | 512 | 512 | 512 | 512 | 512 | 512 | 512 |
| Lily_s | 0.1 | 0.1 | 0.1 | 0.01 | 0.01 | 0.01 | 0 | 0 |
| $ne\_1$ | 2 | 3 | 2 | 4 | 2 | 2 | 0 | 0 |
| $ne\_2$ | 2 | 3 | 2 | 4 | 2 | 2 | 0 | 0 |
| Batch Size | 64 | 32 | 50 | 64 | 32 | 32 | 0 | 0 |

simplicity and enhanced parameter efficiency. By sharing the LP, we eliminate the need to set a separate LP for each layer, thereby reducing the overall parameter count.

Our decision to share the LP is based on the observation of overall redundancy among layers. Specifically, different layers have varying levels of importance (Zhang et al., 2023), and some less important layers do not require a dedicated LP. By not setting a separate LP for these layers, we avoid introducing extra parameter overhead while having a negligible impact on performance. To test that whether sharing LP results in inferior performance, we conduct experiments with no LP sharing on VTAB-1K. The results are shown in Table 15, from which we can observe that the best overall performance (77.3%) is the same as that in the LP-sharing setting. This indicates that even if we employ one LP for each layer, the performance gain is negligible and many of the parameters are actually redundant. However, not sharing LPs results in extra parameter overhead and damages the parameter-efficiency of Lily. Therefore, LP-sharing is a great strategy to eliminate redundancy among LPs and boost the parameter-efficiency of Lily.

# D  WHERE TO APPLY LILY IN TRANSFORMERS?

PEFT methods have been predominantly explored on the Transformer architecture, which consists of multi-head self-attention (MHSA) and multi-layer perceptron (MLP) as its core modules. In this section, we analyze the impact of fine-tuned modules on performance using Lily. Specifically, we compare Lily's performance on the VTAB-1K benchmark under four settings:

- Applying Lily solely to the query and value transformation module in MHSA (denoted as "qv").

- Applying Lily solely to the MLP module (denoted as "mlp").

- Applying Lily to both the query and value transformation module in MHSA and the MLP module (denoted as "qvmlp").

Table 11: Hyperparameter configuration for Lily on VTAB-1K benchmark.

|  | Vision Transformer | Vision Mamba |
|---|---|---|
| Optimizer | AdamW | AdamW |
| Batch Size | 64 | 64 |
| Learning Rate | 1E-3 | 1E-2 |
| Weight Decay | 1E-4 | 1E-3 |
| # Epochs | 100 | 100 |
| LR Decay | cosine | cosine |

Table 12: Details of the datasets used in our commonsense reasoning tasks.

| Benchmark | Description | # Test Questions |
|---|---|---|
| ARC-c | Multiple-choice science | 2376 |
| ARC-e | Multiple-choice science | 1172 |
| OBQA | Multi-step reasoning | 500 |
| SIQA | Social implications | 1954 |
| WinoG | Fill-in-a-blank | 1267 |
| PIQA | Physical commonsense | 1830 |
| BoolQ | Yes/no questions | 3270 |
| HellaS | Commonsense NLI | 10042 |

Table 13: Information about datasets in the GLUE benchmark, with STS-B being a regression task and all other tasks falling into the categories of single-sentence or sentence-pair classification.

| Corpus | Metrics | Task | # Train | # Val | # Test | # Labels |
|---|---|---|---|---|---|---|
| | | Single-Sentence Tasks | | | | |
| CoLA | Matthews Corr. | Acceptability | 8.55k | 1.04k | 1.06k | 2 |
| SST-2 | Accuracy | Sentiment | 67.3k | 872 | 1.82k | 2 |
| | | Similarity and Paraphrase Tasks | | | | |
| MRPC | Accuracy/F1 | Paraphrase | 3.67k | 408 | 1.73k | 2 |
| STS-B | Pearson/Spearman Corr. | Sentence similarity | 5.75k | 1.5k | 1.38k | 1 |
| QQP | Accuracy/F1 | Paraphrase | 364k | 40.4k | 391k | 2 |
| | | Inference Tasks | | | | |
| MNLI | Accuracy | NLI | 393k | 19.65k | 19.65k | 3 |
| QNLI | Accuracy | QA/NLI | 105k | 5.46k | 5.46k | 2 |
| RTE | Accuracy | NLI | 2.49k | 277 | 3k | 2 |

- Applying Lily to both the key and value transformation module in MHSA and the MLP module (denoted as "kvmlp").

To ensure a fair comparison, we tune the hyperparameters to maintain a similar parameter count across all settings. Additionally, to further investigate whether sharing the low-rank projection (LP) affects performance, we do not share the LP in this experiment. The results are presented in Table 15. We observe that the "kvmlp" setting achieves the best performance, with an average accuracy of 77.3%. In contrast, adapting only the MHSA module ("qv") yields the worst performance. Furthermore, we note that adapting both the MHSA and MLP modules (qvmlp and kvmlp) generally leads to superior results compared to adapting only one specific module (qv and mlp). This suggests that both MLP and MHSA play crucial roles in the overall model performance, and adapting both is essential for effective adaptation.

Notably, even when applying Lily solely to the MHSA module, which results in the worst performance among the four settings (76.9%), it still outperforms LoRA by a significant margin (0.5%). This underscores the efficiency of Lily, as it uses fewer parameters than LoRA even without LP sharing.

Table 14: Detailed information about the datasets in VTAB-1K benchmark.

|  | Dataset | Train | Val | Test | #Classes |
|---|---|---|---|---|---|
|  | CIFAR100 |  |  | 10,000 | 100 |
|  | Caltech101 |  |  | 6,084 | 102 |
|  | DTD |  |  | 1,880 | 47 |
|  | Oxford-Flowers102 |  |  | 6,149 | 102 |
|  | Oxford-Pets |  |  | 3,669 | 37 |
|  | SVHN |  |  | 26,032 | 10 |
|  | Sun397 |  |  | 21,750 | 397 |
|  | Patch Camelyon |  |  | 32,768 | 2 |
|  | EuroSAT |  |  | 5,400 | 10 |
| VTAB-1k | Resisc45 | 800/1,000 | 200 | 6,300 | 45 |
|  | Retinopathy |  |  | 42,670 | 5 |
|  | Clevr/count |  |  | 15,000 | 8 |
|  | Clevr/distance |  |  | 15,000 | 6 |
|  | DMLab |  |  | 22,735 | 6 |
|  | KITTI-Dist |  |  | 711 | 4 |
|  | dSprites/location |  |  | 73,728 | 16 |
|  | dSprites/orientation |  |  | 73,728 | 16 |
|  | SmallNORB/azimuth |  |  | 12,150 | 18 |
|  | SmallNORB/elevation |  |  | 12,150 | 18 |

Table 15: Performance on VTAB-1K benchmark when applying Lily to various modules in Transformer. The implementation here does not share LP for simplicity (i.e., each layer has one LP).

|  | Average | Natural | | | | | | | Specialized | | | | Structured | | | | | | | |
|---|---|---|---|---|---|---|---|---|---|---|---|---|---|---|---|---|---|---|---|---|
|  |  | Cifar100 | Caltech101 | DTD | Flowers102 | Pets | SVHN | Sun397 | Camelyon | EuroSAT | Resisc45 | Retinopathy | Clevr-Count | Clevr-Dist | DMLab | KITTI-Dist | dSpr-Loc | dSpr-Ori | sNORB-Azim | sNORB-Ele |
| qv | 76.9 | 73.2 | 92.3 | 72.2 | 99.3 | 91.4 | 89.0 | 56.5 | 87.6 | 95.2 | 84.8 | 75.9 | 83.7 | 65.8 | 52.8 | 81.2 | 87.6 | 52.4 | 36.3 | 43.4 |
| mlp | 77.0 | 74.0 | 92.6 | 72.2 | 99.4 | 91.5 | 89.0 | 55.9 | 88.2 | 95.5 | 85.4 | 76.0 | 83.3 | 63.1 | 53.0 | 81.4 | 86.5 | 53.8 | 35.6 | 43.3 |
| qvmlp | 77.1 | 73.9 | 93.2 | 72.7 | 99.4 | 91.6 | 89.7 | 56.5 | 87.9 | 95.3 | 85.0 | 76.1 | 84.6 | 65.2 | 53.0 | 82.1 | 86.7 | 53.0 | 36.0 | 42.8 |
| kvmlp | 77.3 | 74.0 | 92.3 | 72.6 | 99.3 | 91.5 | 89.2 | 56.7 | 88.2 | 95.4 | 85.3 | 76.0 | 84.6 | 64.9 | 53.4 | 81.7 | 87.5 | 52.9 | 36.9 | 45.2 |

# E  WHERE TO APPLY LILY IN MAMBA?

Nearly all previous PEFT method studies have been centered around Transformers, while Mamba is a relatively new architecture, so there has been little research on PEFT methods on Mamba. In this section, we briefly analyze the pros and cons of adapting Mamba's modules. In brief, a Mamba block consists of regular linear projection layers and a core component SSM module (Gu & Dao, 2023) (Zhu et al., 2024). Specifically in SSM module, Mamba utilize parameters ($\Delta$, $A$, $B$, $C$) to transform an input sequence $x(t)$ to an output sequence $y(t)$ using a hidden state $h(t)$. The discretization process converts $A$ and $B$ into $\bar{A}$ and $\bar{B}$, respectively, using the time step size parameter $\Delta$. Structured state space models, inspired by continuous systems, can be computed similarly to RNNs or in the form of global convolution due to their linear time invariance (LTI) property. Mamba introduces a selective property to structured state space model, tying parameters to the current input, which breaks the LTI property and hinders parallel training. To address this, Mamba employs a hardware-aware algorithm, enabling its SSM module to possess the selective property and perform parallel training. To be specific, the discretization process can be expressed as:

$$\bar{A} = exp(\Delta A)$$
$$\bar{B} = (\Delta A)^{-1}(exp(\Delta A) - I) \cdot \Delta B \tag{7}$$

After that, the calculation in Mamba can be expressed as:

$$h_t = \bar{A}h_{t-1} + \bar{B}x_t$$
$$y_t = Ch_t \tag{8}$$

Table 16: Commonsense reasoning results of Lily under various leanring rates.

| Model | Lr | BoolQ | PIQA | SIQA | HellaSwag | WinoGrande | ARC-e | ARC-c | OBQA | Avg. |
|-------|-----|-------|------|------|-----------|------------|-------|-------|------|------|
| | 1e-3 | 70.7 | 84.6 | 77.6 | 87.8 | 77.3 | 88.5 | 74.1 | 80.8 | 80.2 |
| LLaMA3-8B | 5e-4 | 71.8 | 86.5 | 77.9 | 82.8 | 83.1 | 88.6 | 76.8 | 81.4 | 81.1 |
| | 3e-4 | 72.9 | 85.6 | 77.8 | 92.7 | 83.3 | 89.7 | 77.6 | 82.8 | 82.8 |

```python
class lily_adapter_monoscale(nn.Module):

    def __init__(self, hidden_dim, ne, lp, hps, mlp=False):
        super().__init__()
        self.hps = hps
        self.ne = ne
        self.lp = lp
        self.scale = 1 / ne
        if mlp:
            self.non_linear = nn.ReLU()
        else:
            self.non_linear = nn.Identity()
    def forward(self, x):
        hidden = self.non_linear(self.lp(x))
        combined_hp = torch.sum(self.hps, 0) * self.scale
        return torch.matmul(hidden, combined_hp)
```

Figure 10: Implementation of Lily with no selectivity.

where $h_t$ is the hidden state at time $t$ and $x_t$ is the corresponding input token. Delta projection is a module in SSM that's learnable and tasked with transforming the parameter $\Delta$. Since adapting the delta projection alone can indirectly adapt the entire SSM module (i.e., $\bar{A}$ and $\bar{B}$ are all determined by $\Delta$), it is the most critical component of the SSM module.

We investigate the performance of two adaptation strategies: adapting only the input linear projection layer (denoted as "in") and adapting both the input linear projection layer and SSM (denoted as "$\Delta$ + in" since we only adapt delta projection in SSM module). Our results, as shown in Table 1, indicate that applying Lily solely to the input projection yields better performance than applying it to both the input and delta projection modules. This suggests that when adapting Mamba-based models under the paradigm of low-rank adaptation, it is optimal to adapt only the input projection module outside the SSM module. These findings highlight the need for further research into the impact of fine-tuned modules in Mamba on overall performance. Additionally, developing PEFT methods specifically tailored to Mamba-based models, whether for vision or language foundation models, is also a promising direction for future work.

## F  PERFORMANCE WITH DIFFERENT LEARNING RATES

Since we only tuned the learning rate in the commonsense reasoning experiment, we provide the performance of commonsense reasoning under different learning rates in Table 16.

## G  DOES SELECTIVITY HELP?

Lily introduced selective weight combination to selectively incorporate information from other layers. To verify the effectiveness of this selectivity, we remove the router from Lily and evaluate the impact. The modified algorithm without the router is presented in Fig. 10. We conduct experiments on commonsense reasoning to investigate the effect of removing selectivity from Lily.

As shown in Table 17, removing selectivity from Lily results in generally poorer performance compared to vanilla Lily. This is likely because the lack of selectivity causes Lily to simply aggregate all the HP expert, leading to inferior performance. This validates the design choice of using routers in Lily to selectively allocate weights to HP experts, rather than simply summing them.

Table 17: Commonsense reasoning results of Lily without selectivity. We provide results using two learning rates.

| Model | Lr | BoolQ | PIQA | SIQA | HellaSwag | WinoGrande | ARC-e | ARC-c | OBQA | Avg. |
|-------|-----|-------|------|------|-----------|------------|-------|-------|------|------|
| | 3e-4 | 64.0 | 82.6 | 78.5 | 77.0 | 79.6 | 88.4 | 74.5 | 82.0 | 78.3 |
| LLaMA3-8B | 5e-4 | 71.3 | 85.5 | 78.1 | 84.3 | 79.6 | 86.4 | 76.1 | 79.0 | 79.8 |

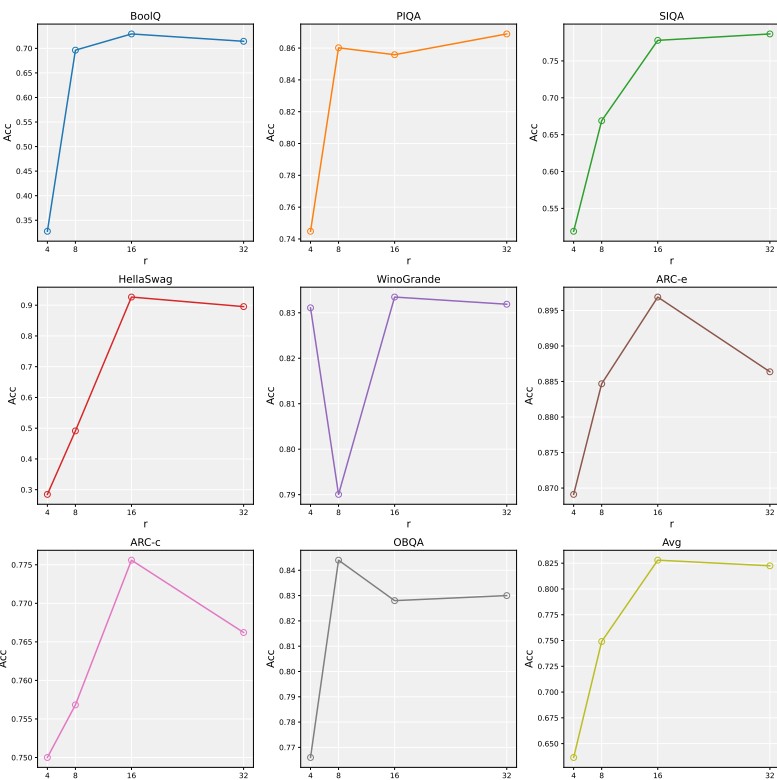

Figure 11: Results on commonsense reasoning tasks when applying different settings of rank. The hyperparameter *ne* is specifically tuned to maintain the same amount of parameter count for a fair comparison.

## H   HOW TO ALLOCATE PARAMETERS?

Since Lily alters the traditional LoRA's layer-bound setup, increasing the parameters of Lily can be achieved through two approaches: 1) increasing *ne*, i.e., increasing the number of LP and HP experts, and 2) increasing the rank, i.e., increasing the parameter size of each individual LP or HP expert. In this section, we investigate which factor has the greatest impact on performance. We conduct experiments on the commonsense reasoning task. Specifically, we maintain the same parameter count and learning rate, and achieve the same parameter count by setting different ranks and adjusting the corresponding *ne* (e.g., r=16, *ne*=4 versus r=8, *ne*=8). The results are shown in Fig. 11, from which we observe that more LP and HP experts with smaller rank (i.e., bigger *ne* and smaller rank) generally performs worse. We argue that this is because, although increasing the attention granularity allows for finer details, the resulting performance gain is not as significant as the gain obtained by increasing the rank, i.e., increasing the model's capacity to learn more information. This gives us an insight that, in Lily, increasing *ne* to increase the parameters is less effective than directly increasing the rank in terms of potential performance gain.

# I    MORE ON SUBJECT-DRIVEN GENERATION

We provide more results on subject-driven generation in Fig. 12 and Fig. 13.

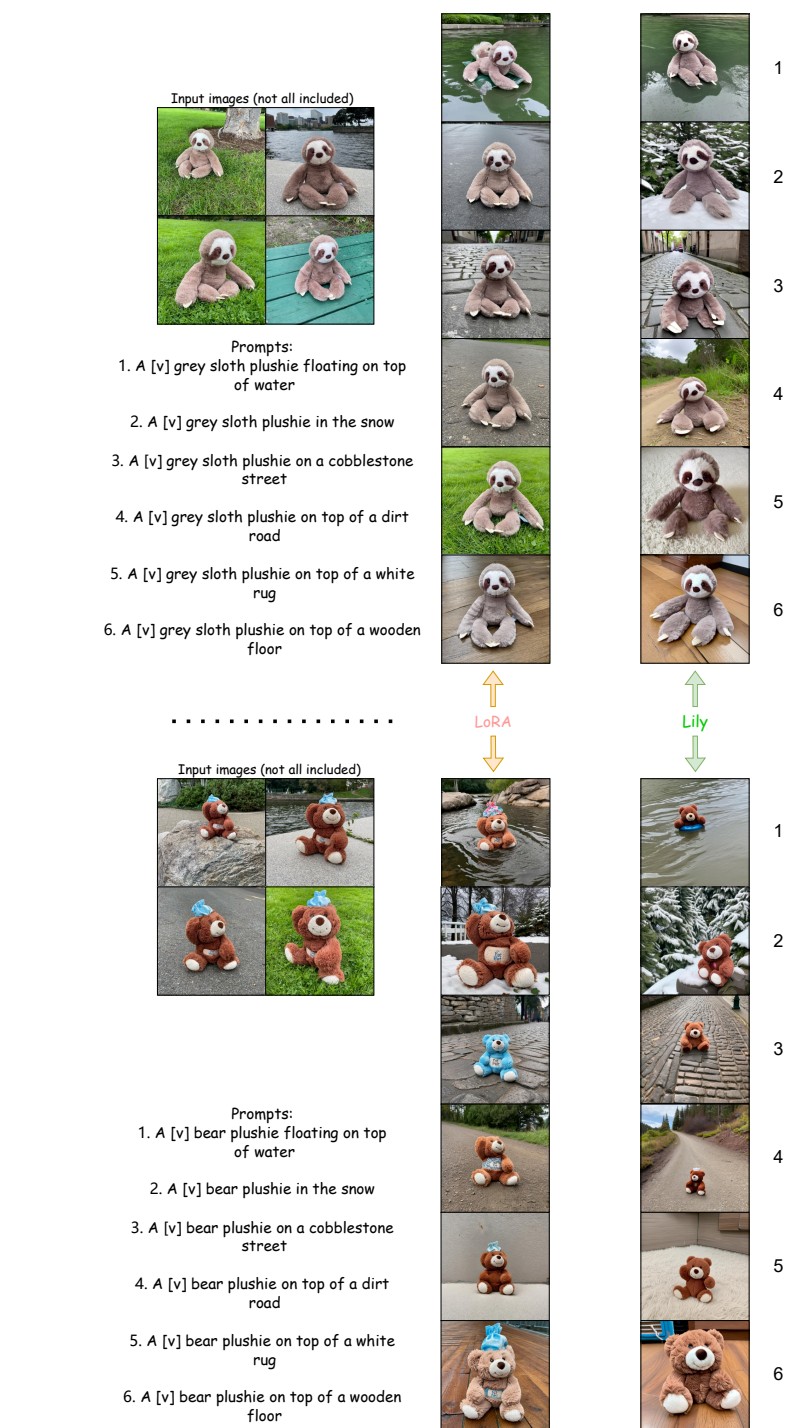

Figure 12: More subject-driven generation results for unreported subjects.

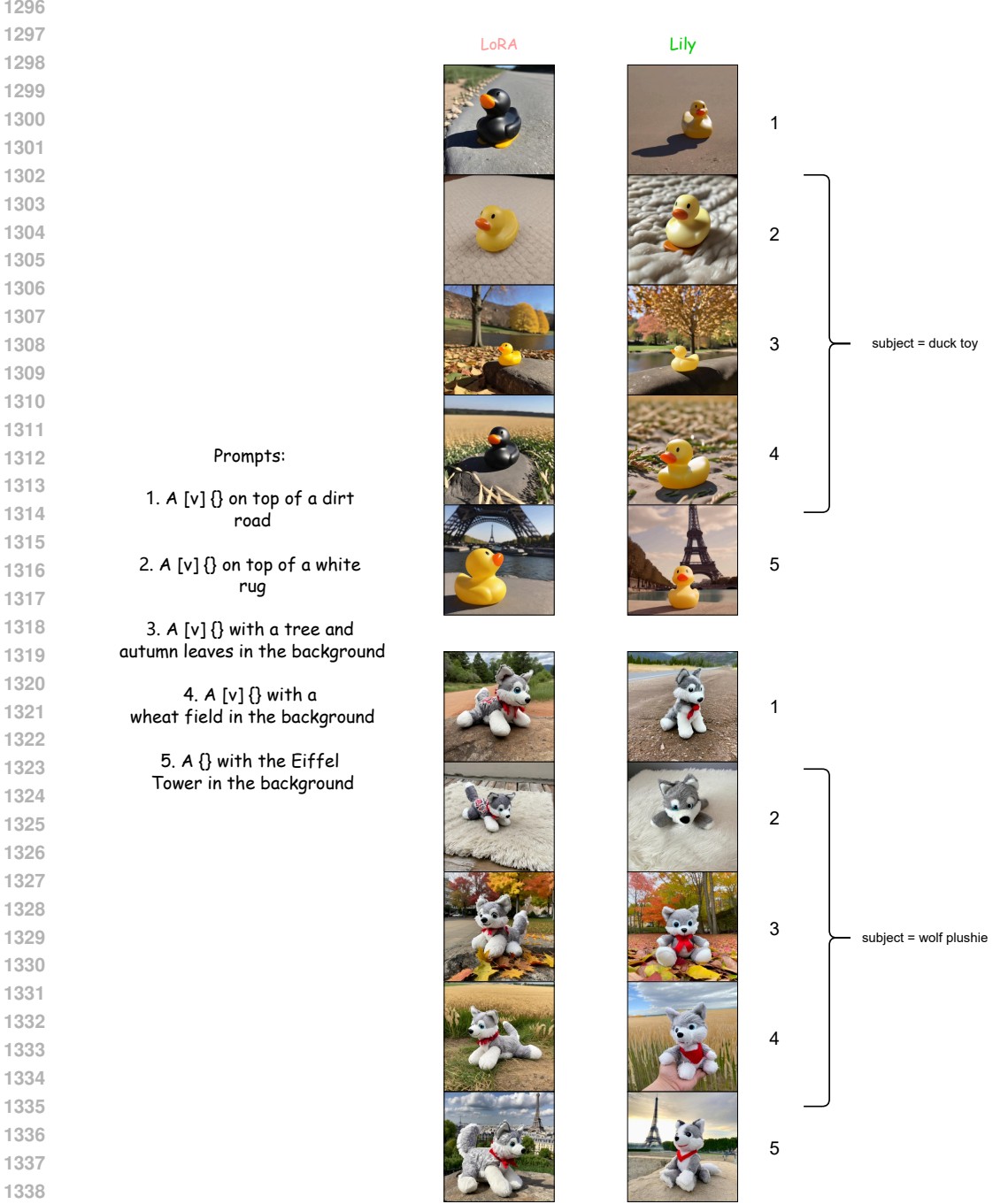

Prompts:

1. A [v] {} on top of a dirt road

2. A [v] {} on top of a white rug

3. A [v] {} with a tree and autumn leaves in the background

4. A [v] {} with a wheat field in the background

5. A {} with the Eiffel Tower in the background

Figure 13: More subject-driven generation results for subjects that are reported in the experiment section.

## J MORE ON ATTENTION MAPS OF LILY AND LORA

We provide more visualization results of the attention map from both LoRA and Lily on Caltech101 dataset from VTAB-1K benchmark in Fig. 14.

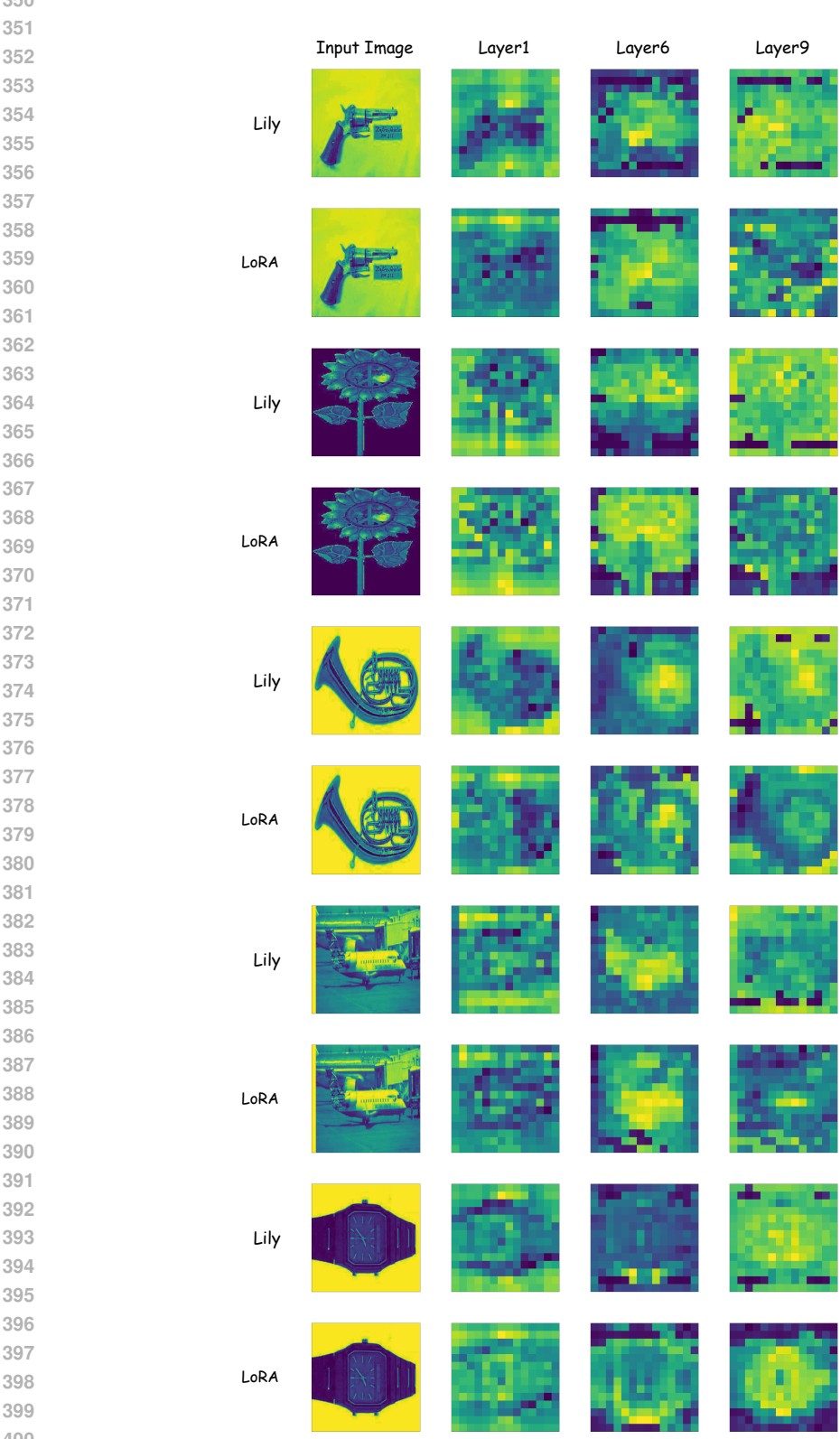

Figure 14: More results of attention maps from LoRA and Lily. All images are taken from Caltech101 dataset.

