# OpenReview forum: "Low-Rank Interconnected Adaptation across Layers"
_ICLR.cc/2025/Conference — Submitted to ICLR 2025_

### Official Review · Reviewer_dXjD · 2024-10-22

**Soundness:** 2
**Presentation:** 2
**Contribution:** 3
**Rating:** 5
**Confidence:** 4

**Summary:**

The Authors propose an alternative ansatz to LoRa for parameter efficient finetunig of neural networks. In particular, they propose to connect all low-rank adapters of a given network to each layer,  combined with a router. This way a weighted sum of low-rank adapters is modifying the parameters of a given layer.
The authors demonstrate the method in an arrray of contemporary benchmarks.

**Strengths:**

I find the idea of using a sum of low-rank adapters interesting.
The ansatz potentially enables a high rank adaptation using a sum of low-rank adapters.

The method is demonstrated at an impressive number of numerical tests.

**Weaknesses:**

While the general idea is promising and supported by numerical evidence, I find the presentation of the method description and theoretical premisses thereof lacking in soundness and readbility. In my opinion the first part of the paper requires a major revision before being publishable.

In particular:

- The work is based upon the premise that the B matrix is very close to 0 or equals to zero. This claim has been investigated by Hao et al. however under strong assumptions about the zero initialization of B and very small learning rates and only for the "classical" LoRa setup.
The authors of this work just use the conclusions of Hao's analysis as the premise for their method. Two points are critical here
 1) The assumptions in the LoRa setting are not well communicated in both Introduction (Section 1) and Section 3.1. In Section 3.1 the mathematical notation is very sloppy, using equality signs between eq. 2 and 3, when this is in reality a first order approximation for small learning rates and values of B. This is missleading, especially for readers not familiar with Hao's work. The same applies for eq. 9 and 10.

2) The validity of the assumptions need to be shown for the newly introduced ansatz and method of this paper, instead of just taken for granted.

- The authors claim that their ansatz (eq. 14). allows high rank updates with low-rank matrices. While it is generally true, that a sum of (even) rank 1 matrices can be high rank, it does not have to be the case. An (very simple) numerical experiment is needed to demonstrate the effectivity of this ansatz. One could think of a matrix regression problem with said ansatz. A stage 2 numerical experiment would be the anlysis of the effective ranks of each layer of an adaption of the proposed kind.
The same ciritique is applicable for the MoE ansatz in the paper.

- The method should be illustrated with an algorithm.

- The effective computational efford of the method needs to be analyzed. Having strong interconnections between layers significantly increases the complexity of the gradient tape, which increases compute and memory consumption of the method, potentially defeating the purpose of PEFT.

- Some statements are quite vague, e.g. Line 226: "shallow and deep inputs" what does that mean?

**Questions:**

- Does the method require that exactly L (where L is the number of layers) low-rank adapters need to be combined? I think the notation can be simplified by defining the number of adapters independently of layers, especially for the MoE ansatz.

- Have you measured the effictive compute batch training time in comparison with adalora or lora?
- Can your method be extended to be rank adaptive?
- How does the norm of B change during training? Your method is based upon the assumption that the values of B are very small. Is this still the case after several epochs? In this regard: What are the practical thresholds for the learning rate, where the method assumptions breal?

---

> ### Author Response · Authors · 2024-11-17
> **Replies to Reviewer dXjD: Part 1**
>
> We greatly appreciate your professional and insightful feedback! Here we provide a detailed response to all of your concerns below. One important point to note is that **we will remove the theoretical analysis of the rank (section 3.1) from the main text** and leave it to a section in the appendix for the convenience of the readers who favor some mathematical analysis in our revised version. This is because the current analysis is based on strong assumptions as you point out and lacks rigourous proof as reviwer F3Lx also points out, which is sadly beyond our current expertise :(
>
> **However, as we will demonstrate with data and experiments in our reply to your Q2, our claim that Lily achieves high-rank updates with low-rank structure is true and validated by experiments. We will also provide this numerical and emprical analysis instead of the mathematical one in our main text and it is actually much more convincing than our current theoretical analysis :)**
>
>
> ---
>
> ## Q1(1): The assumptions about the zero initialization of B and very small learning rates are too strong and only for the "classical" LoRA setup
>
> Thanks for your feedback. The practice of initializing $B$ with an all-zero matrix and $A$ with a normal distribution is so common in the deployment of LoRA that we believe this particular assumption is practical and reflects real-world conditions. About the issue of small learning rates, please refer to our reply to Q9, where we discuss it more deeply with actual data. In this case, the assumptions are indeed way too strong and we therefore opt for empirical validation for our method.
>
> As for the setup is only for classical LoRA, we want to highlight that despite multiple works are being proposed about the initialization aspect of LoRA, classical LoRA is still one of the state-of-the-art PEFT methods that is simple, efficient and widely supported and deployed. We therefore believe that basing our work on classical LoRA setup is both realistic and practical.
>
> ---
>
> ## Q1(2): The assumptions in the LoRa setting are not well communicated in both Introduction (Section 1) and Section 3.1
>
> Thanks for pointing this out. We will make changes accrodingly in the revised version to ensure the assumptions is well informed to the readers!
>
>
> ---
> ## Q1(3): using equality signs between eq. 2 and 3, when this is in reality a first order approximation for small learning rates and values of B
>
> Eq 2 is $W = W_0 + (A_0+\Delta A)(B_0 + \Delta B)$, which after expanding is: $W = W_0 + A_0B_0 + A_0\Delta B + \Delta AB_0 + \Delta A\Delta B$. Since we state before Eq. 2 that "$B_0$ is initialized as 0", the term $A_0B_0$ and $\Delta AB_0$ are in fact zero. Therefore, deliminating these two terms makes the following equation: $W = W_0+A_0\Delta B+\Delta A\Delta B$, which is Eq 3 in the paper, and we think the usage of equality signs is correct here.
>
> We think what you mean by "this is in reality a first order approximation for small learning rates and values of B" is refering to **Eq.3 to Eq. 4 and Eq. 10 to Eq. 11**. Because using the approximation based on small learning rate, the term $\Delta A\Delta B$ in Eq. 3 can then be approximately eliminated. **Note that we do use "approximately equal to" from Eq.3 to Eq. 4 and Eq. 10 to Eq. 11.**
>
> ---
>
> ## Q1(4): The validity of the assumptions need to be shown for the newly introduced ansatz and method of this paper, instead of just taken for granted.
>
> Thanks for pointing this out, please refer to our reply to Q9 for a detailed and numerical study of this issue! **We basically recognise the validity of the assumptions is not well-established and we opt for an empirical validation of our claim instead.**

---

> ### Author Response · Authors · 2024-11-17
> **Replies to Reviewer dXjD: Part 2**
>
> ## Q2: provide numerical experiment to demonstrate the effectivity of this ansatz
>
> Thanks for the suggestions! We analyse the ranks of the weight updates for each target after adapting RoBERTa-Base in MRPC dataset below. We run 20 epochs and use 2LPs and 2HPs for Lily, the same configuration in our paper. For simplicity, we consider $W_q$ in layer 1, 2 and 3. However, this trend holds in other layers and other targets as well. For LoRA we use a hidden dimension of 8 and for Lily we use 32 and 8, respectively. Note that the weight update is of shape $[768, 768]$ here. We implement this analysis using torch.linalg.matrix_rank(matrix).
>
> | method      | param | layer1 | layer2 | layer3 |
> | ----------- | ----- | ------ | ------ | ------ |
> | LoRA   | 0.3M  | 27     | 29     | 24     |
> | Lily | 0.3M  | 224    | 225    | 222    |
> | Lily  | 0.05M | 75     | 76     | 77     |
>
> Therefore, it is important to note that the rank of **Lily's weight updates is significantly larger than LoRA's when using the same amount of parameters**, with only 2LPs and 2HPs. When decreasing the hidden dimension of Lily from 32 to 8, the **parameter count of Lily is only $16.7\%$ of LoRA's, while still achieving roughly $2.8\%$ higher rank in terms of the actual weight updates**. This empirical study effectively **validate our claim that Lily achieves high-rank weight updates despite using a low-rank structure.**
>
> In terms of the MoE ansatz, we actually provide a **visualization of the activity of the router in Figure 6**, where we show the weights it allocates to the HP experts. However, we do provide a **table of numerical value** of the cifar100 dataset in Figure 6 of the paper when  down below. Each cell contains the **accumulative** weight (or importance score) given by the router from that layer to the expert.
>
> |         | layer2     | layer13     | layer22    |
> | ------- | ---------- | ----------- | ---------- |
> | expert1 | 6868221.5  | 1106866.625 | 6979373.2  |
> | expert2 | 1177594.25 | 9600196.3   | 3833344.5  |
> | expert3 | 3550391.75 | 876543.5    | 770883.125 |
>
> It can be observed that the introduction of router, inspired from MoE, effectively allocate different weight to each experts based on feature from the current layer. This process is dynamic and therefore making the adaptation process more expressive. We provide further analysis of the MoE concept of Lily in secion 4.5.
>
> ---
>
> ## Q3: The method should be illustrated with an algorithm.
>
> We have put the actual code implemented in PyTorch in Figure 7. However, we will add an algorithm in the revised version.
>
> ---
>
> ## Q4: The effective computational efford of the method needs to be analyzed
>
> Here are an analysis of Lily versus LoRA and AdaLoRA in terms of speed and memory. We run the analysis on QNLI from GELU tasks. We use 0.3M parameters and run 1 epoch to test the speed efficiency. We will add this in our revised version.
>
> | method  | time (sec) | memory   | params |
> | ------- | ---------- | -------- | ------ |
> | LoRA    | 145.299978 | 18.40GiB | 0.3M   |
> | AdaLoRA | 157.636769 | 18.42GiB | 0.3M   |
> | Lily    | 150.342264 | 20.51GiB | 0.3M   |
>
> It can be obsered that while Lily does introduce a certain level of complexity to the gradient computation, makeing it's memory comsumption slightly bigger than LoRA and AdaLoRA. In terms of the speed, Lily is slightly slower than LoRA and slightly faster than AdaLoRA. However, the actual speed and memory is mostly comparable to those of LoRA and AdaLoRA. Additionally, considering the fact that a large part of the downstream adaptation using PEFT is performed on small-scale datasets and models, and only a small fraction of the parameters are tuned, this small distinction in speed and memory usage is further negligible. **Therefore, we believe Lily is a pretty effective PEFT method in terms of hardware-friendliness despite its additional complexity.** That is to say, Lily's introduced complexity over LoRA does not prevent it from an decent PEFT method in terms of training speed and memory.
>
> ---
>
> ## Q5: Some statements are quite vague, e.g. Line 226: "shallow and deep inputs" what does that mean?
>
> We are very sorry about the confusion! What we mean is inputs to shallow layers in the model (say layer number 1 or 2) and inputs to deep layers in the model (say layer number 9 or 10 in a 12-layer model). We will clarify this in our revised version.
>
> ---
>
> ## Q6 : Does the method require that exactly L low-rank adapters need to be combined?
>
> No, actually not at all! We have denote the number of LPs of HPs using $N_l$ and $N_e$ in the methodology part. In the experiment, we denote number of LPs by simply $ne\_1$ (ne stands for number of experts) and number of HPs by $ne\_2$. $ne\_1$ and $ne\_2$ can all be flexibly adjusted to enable maximum parameter efficiency. **In fact, we do not combine exactly L adapters in our experiments (we discuss this in Appendix A), rather using much lesser than L adapters to boost the parameter-efficiency.**

---

> ### Author Response · Authors · 2024-11-17
> **Replies to Reviewer dXjD: Part 3**
>
> ## Q7:  Have you measured the effective compute batch training time in comparison with adalora or lora?
>
> Please refer to our reply to Q4!
>
> ---
>
> ## Q8: Can your method be extended to be rank adaptive?
>
> This could be an interesting direction for our future work. AdaLoRA mainly addresses the problem of evenly distributing the budget of incremental updates across all pre-trained weight matrices, like in LoRA. The intuition is that matrices across the layers are not equally important, some layers have higher importance while other have less. **In Lily, since the HPs ($B$ in LoRA) are model-wide shared and the interconnectivity of LPs and HPs, it means that basically matrices across the layers are actually in some degree getting the benefit from all of the parameter budget, but with varying dynamics and weights determined by the MoE router.**
>
> Therefore, Lily actually partially address the issues which AdaLoRA addresses (every one gets all the budget by model-wide sharing and interconnectivity). However, it will be promising to further considering this direction of research, thank you for your advice!
>
> ---
>
> ## Q9: How does the norm of B change during training? Your method is based upon the assumption that the values of B are very small. Is this still the case after several epochs? In this regard: What are the practical thresholds for the learning rate, where the method assumptions breal?
>
> TL;DR: we do find that these assumptions are way too strong and can not reflect real world scenarios! **However, as we stated before, we replace the theoretical analysis with a much stronger and more convincing empirical analysis.**
>
> We here provide an analysis of the norms in layer 1 for $W_q$ below. Note that we use zero initialization of B, therefore B equals $\Delta B$. The dataset we use is MRPC.
>
> ### learning rate = 5e-5: (very small for PEFT)
> In this case the assumptions are reasonable. $\Delta A \Delta B$ does can be ignored.
>
> | component          | epoch=5 | epoch=10 | epoch=20 |
> | ------------------ | ------- | -------- | -------- |
> | $\Delta B$         | 0.0514  | 0.0705   | 0.1083   |
> | $\Delta A$         | 0.0599  | 0.0613   | 0.1107   |
> | $\Delta A\Delta B$ | 0.0014  | 0.0024   | 0.0068   |
> | $A_0\Delta B$      | 0.0298  | 0.0405   | 0.0621   |
> ### learning rate = 5e-4: (typicaly for adapting LLMs, not small models)
> In this case the assumptions are trickier. $\Delta A \Delta B$ seems to be comparable to $A_0 \Delta B$, which can not be ignored.
>
> | component          | epoch=5 | epoch=10 | epoch=20 |
> | ------------------ | ------- | -------- | -------- |
> | $\Delta B$         | 0.5357  | 0.8539   | 1.2623   |
> | $\Delta A$         | 0.5759  | 0.9445   | 1.6066   |
> | $\Delta A\Delta B$ | 0.1719  | 0.3962   | 1.8136   |
> | $A_0\Delta B$      | 0.3109  | 0.4959   | 0.7205   |
>
> ### learning rate = 5e-3: (typicaly for adapting small models, not LLMs)
> In this case the assumptions are wrong. The term $\Delta A \Delta B$ actually is bigger than $A_0 \Delta B$.
>
> | component          | epoch=5 | epoch=10 | epoch=20 |
> | ------------------ | ------- | -------- | -------- |
> | $\Delta B$         | 3.4694  | 4.1968   | 5.7798   |
> | $\Delta A$         | 3.9257  | 4.5037   | 5.8040   |
> | $\Delta A\Delta B$ | 9.9233  | 12.7000  | 27.5966  |
> | $A_0\Delta B$      | 2.0048  | 2.4412   | 3.2891   |
>
>
> We believe this analysis validate your concerns that the assumptions are way too strong, and **we opt for a stronger and more convincing experimental study of our claim instead, as shown in Q2!**
>
>
> ---
>
> If you still have concerns, please reply to us and we will try our best to address all of them! We want to kindly restate that although our theoretical analysis is no longer used and we will not show them in the main text, replacing it with empirical analysis actually strengthen our claim :)

---

> ### Author Response · Authors · 2024-11-21
>
> Dear Reviewer dXjD:
>
> We thank you again for your valuable feedback! Our responses were posted four days ago, but it appears that none of the reviewers have provided further thoughts or questions. It seems that you may have missed the notification about our responses. If that is the case, we would like to hear from you and are open to any further questions. We will address all of the concerns and improve our paper according to your feedback!

---

> > ### Author Response · Authors · 2024-11-27
> >
> > Dear Reviewer dXjD,
> >
> > Thank you once again for your constructive feedback. As the discussion period is drawing to a close, we would appreciate hearing from you regarding our responses. We have revised the paper according to your feedback, for example, by including an empirical analysis of the rank gain of Lily compared to LoRA, and we hope your concerns have been properly addressed.
> >
> > We look forward to your thoughts and hope to bring this review process to a successful conclusion.

---

> ### Author Response · Authors · 2024-12-01
> **Please reply to us and consider raising your score**
>
> Dear reviewer dXjD,
>
> Thank you so much for your precious time and effort. As the discussion period is coming to its end, could you kindly read our responses and reply to us? If we have successfully addressed your concerns, we respectfully ask that you consider raising your score accordingly.
>
> Thank you again for your attention and effort!

---

### Official Review · Reviewer_F3Lx · 2024-10-31

**Soundness:** 2
**Presentation:** 1
**Contribution:** 3
**Rating:** 3
**Confidence:** 4

**Summary:**

This paper investigates the limitation of weight update in LoRA the limitations of weight updates in LoRA, highlighting that it functions as a random projection on an initialized low-rank matrix. To address such limitations, the author introduces low-rank interconnected adaptation across layers (Lily), which employs model-wise shared low-rank matrices inspired by the mixture of experts (MoE) framework. Extensive experiments have been conducted to demonstrate the effectiveness and adaptability of the proposed methods.

**Strengths:**

1. The motivation is reasonable as learning multiple shared low-rank matrices can represent various low-rank subspaces, and their combinations have the potential to capture information across a higher-rank space.

1. Comprehensive experiments demonstrate the effcacy and adaptability of Lily, achieving state-of-the-art results across a diverse set of tasks in both language and vision domains.

**Weaknesses:**

1. The presentation of the methodology has significant room for improvement.

    1. The description of the methodologies (lines 204-210) and the framework depicted in Figure 1 within the main text differ from the final implementation discussed in Appendix A.1.1 (lines 770-778). In the main text, low-dimensional projectors (LPs) are described as being tied to each layer of a module, while high-dimensional projectors (HPs) are shared across the model. However, Appendix A.1.1 indicates that Lily essentially learns a set of model-wide shared LPs and HPs that are selectively applied to specific layers.

    1. The derivations from Eqn. 9-14 lack rigor in scenarios involving multiple LPs and HPs. To extend the derivation from Eqn. 5-20 in [1] to multiple LPs and HPs, several adjustments are needed:

         - **First**, Eqn. 9 should incorporate a summation over the LPs, leading to a double summation in Eqn. 11.

         - **Second**, gradient with respect to each $B_{i}$ should be considered.

         -  **Third**, the applicability of Theorem 2.1 in [1] for multiple LPs and HPs must be proven before use.

         Without proper proofs, simple substitutions may lead to inaccuracies, making the current mathematical discussion unconvincing.

    1. Eqn. 17 - 18 are just simple math that does not warrant separate discussion. In addition, the definition of $s$ in Eqn. 19 - 20 is missing.

    1. In essence, it is advisable to directly present the core idea (a set of multiple LPs and HPs) if the theoretical discussion lacks sufficient proof.

1. Concerns regarding experimental results.

    1. The reported number of trainable parameters for Lily is unclear.

        - **First**, Lily is reported to have the same parameter count as LoRA on GLUE tasks while using only 1/4 of the parameters compared to LoRA on ViT tasks. Given that vision tasks are typically more complex than GLUE tasks and generally require more parameters, this discrepancy needs clarification.

        - **Second**, based on the configurations in Tables 9 and 10, Lily only uses 2 LPs and 2 HPs shared across the entire model. Under this setup, the parameter count should be substantially lower than that of LoRA for various backbones, such as RoBERTa-base/ViT-base (12 layers), yet Lily's parameter count appears comparable to LoRA on GLUE. Further explanation is needed to resolve this inconsistency.

    1. Efficiency is a critical aspect of PEFT. However, for the most computationally demanding task, commonsense reasoning, the parameter count is not provided. It would be helpful to include the parameter count and training time for this task, as shown in [2].

    1. The experimental results do not include MNLI and QQP, despite these tasks being listed in the configuration table.

[1] Yongchang Hao, et al. FLORA: Low-Rank Adapters Are Secretly Gradient Compressors. In ICML 2024.

[2] Taiqiang Wu, et al. Mixture-of-Subspaces in Low-Rank Adaptation. Arxiv.

**Questions:**

See Weakness.

---

> ### Author Response · Authors · 2024-11-17
> **Replies to Reviewer F3Lx: Part 1**
>
> We highly appreciate your insightful and constructive feedback! We provide our detailed responses to all of your concerns down below.
>
> One important point to note is that **we remove the theoretical analysis of the rank (section 3.1) from the main text** and leave it to a section in the appendix for the convenience of the readers who favor some mathematical analysis. This is because the current analysis is based on strong assumptions, as reviwer dXjD points out and lacks rigourous proof as you also points out, which is sadly beyond our current expertise :(
>
> **However, as we will demonstrate with data and experiments in our reply to your Q1(4), our claim that Lily achieves high-rank updates with low-rank structure is true and validated by experiments. We will also provide this numerical and empirical analysis instead of the mathematical one in our main text.** **We find that this empirical analysis could reflect real-world scenarios and generally is much more convincing than our theoretical analysis :)**
>
> ---
>
> ## Q1(1): The description of the methodologies (lines 204-210) and the framework depicted in Figure 1 within the main text differ from the final implementation discussed in Appendix A.1.1 (lines 770-778).
>
> Thanks for pointing this out. This indeed is a point which needs more clarification in the main text. Let's term the description in the methodology section as **Lily v1** (which is exactly the original implementation) and the one in appendix as **Lily v2** (implementation which we uses for the majority of the experiments) for the ease of reference. Our clarification is below.
>
> **In Lily, the LPs are** **not model-wide, they are always fixed to a specific range (whether a range of neighboring layers or just 1 layer)**. Here's an example of the deployments of LP in Lily v1 and v2: suppose we have 9 layers, in Lily v1 there will be 9 LPs deployed to each of the layer. In Lily v2 however, **neighboring layers could share LPs** for the purpose of further reducing the parameter count and enhance efficiency. For example, if 3 LPs are required, layer 1, 2, 3 will use LP1, layer 4, 5, 6 will use LP2 and layer 7, 8, 9 will use LP3. Therefore, **LPs in Lily v2 are still local, like in v1**, deep layers (say the last layer) will not have access to the shallow LP (say LP from the first layer). The only difference of v1 and v2 is **neighboring layers of a certain range share their LPs**.
>
> Additionaly, we actually **provide comprehensive VTAB-1K benchmark results of Lily v1 in Appendix D**, without this LP-sharing. The results indicates that **sharing LPs (Lily v2) not only reduce the parameters but also perform equally well as Lily v1**.
>
> We also want to point out that our method employs an asymmetric structure where for an adaptation target $W$, there's only **one** $A$ for it (whether it's shared in a range of layers or not) and multiple model-wide $B$s for it. Since **Lily v1 basically is a special case of Lily v2**, we use it to illustrate the weight updates of Lily in the main text for simplicity. For v2, the derivation still holds and thus higher upper-bound of the weight updates' ranks.
>
> ---
>
> ## Q1(2): Derivations from Eqn. 9-14 lack rigor in scenarios involving multiple LPs and HPs
>
> Thanks for your feedback! Before providing the responses, we want to highlight that in the updates process, for an adaptation target $W$, there's only **one local** $A$ (an LP) involved, with multiple model-wide $B$s (HPs).
>
> - As stated above, since Eq. 9 is about the weight updates for an adaptation target where only a single LP is used, we believe a summation over the LPs is not needed.
> - The gradient w.r.t. all the $B$s (or HPs) are indeed considered in Eq. 14. To be specific, term $$\sum_{j=1}^{N_l}{C_{i,j}A_0 A_0^j}^T(\nabla_{W_j}\mathcal{L}_t)$$
>
> in Eq. 14 is the gradient w.r.t a single $B$, **while the summation of these gradient in Eq. 14 ($\sum_{i=1}^{N_e}$) is applying the chain rule and effectively considering gradients of all the $B$s.**
> - Thanks for pointing this out. Please see our response to Q1(4)!
>
> ---
>
> ## Q1(3): Eqn. 17 - 18 are just simple math that does not warrant separate discussion. In addition, the definition of s in Eqn. 19 - 20 is missing.
> Thanks for pointing this out, we will remove Eq. 17 - 18 in our revised version and we will add the definition of s in it.

---

> ### Author Response · Authors · 2024-11-17
> **Replies to Reviewer F3Lx: Part 2**
>
> ## Q1(4):  In essence, it is advisable to directly present the core idea (a set of multiple LPs and HPs) if the theoretical discussion lacks sufficient proof.
>
> Thanks for your general feedback. **We will directly present the core idea and leave a more rigorous proof as our future work, as it is sadly beyond our expertise.** We do leave the current theoretical discussion in Appendix for readers who are interested and will change it when we have the ability to rigorously prove it.
>
> Additionally, we want to kindly remind you that LPs are not model-wide shared and this is the main reason of the simplicity of our derivations.
>
> **Here is an empirical analysis of the actual rank of the weight updates when using Lily versus LoRA**. To be specific, we use the MRPC task from the NLU experiments and test the rank of the $W_q$ when running 20 epochs. The results below generalize well to other targets and layers as well. For LoRA we use a hidden dimension of 8 and for Lily we use 32 and 8, respectively. We use only small number of LPs and HPs (2 and 3) in this experiment for simplicity and to match the parameter count. Note that the weight update is of shape $[768,768]$.
>
> | method      | param | layer1 | layer2 | layer3 |
> | ----------- | ----- | ------ | ------ | ------ |
> | LoRA  | 0.3M  | 27     | 29     | 24     |
> | Lily | 0.3M  | 224    | 225    | 222    |
> | Lily | 0.05M | 75     | 76     | 77     |
>
> It can be observed that the **rank of weight updates using Lily is much higher than using LoRA, under the same parameters count**. Meanwhile, when using $16.7$% of the parameters of LoRA, Lily still achieves roughly $2.8$ times higher rank in terms of the actual weight updates. This empirical study effectively **validate our claim that Lily achieves high-rank weight updates despite using a low-rank structure.**
>
> ---
>
> ## Q2(1)：Discrepancy of the parameter count needs clarification
>
> - The discrepancy in NLU and vision tasks stems from the fact that the cross-layer connection in Lily intuitively will enable stronger visual adaptation capability. We analyse this in section 4.5, that Lily allows feature merging and selective reusing of visual features across different layers, which is ideal for vision tasks. Therefore we were able to outperform LoRA using much reduced parameters. However, in GLUE tasks, Lily can not outperform LoRA when using this-level of reduced parameters (roughy 1/4). Therefore, we opt to use a similar parameter count in GLUE tasks. **Overall, despite being a better PEFT method, the advantage of Lily is more pronounced in visual tasks than in NLU tasks like GLUE tasks.**
> - When using 2 LPs and 2 HPs (actually there are also setups like 3x3 and 4x4 in table 10), **we set the hidden dimension of Lily to 32, while the hidden dimension for LoRA is fixed to 8**. We did this because 1) **ensure the same order of parameter count**. 2) based on our observation in Appendix H (How to Allocate Parameters in Lily?), when **using the same parameter budget, it is advisable to use bigger rank with lesser LPs or HPs** since it will yield better results.

---

> ### Author Response · Authors · 2024-11-17
> **Replies to Reviewer F3Lx: Part 3**
>
> ## Q2(2): Include the parameter count and training time for commonsense reasoning tasks.
>
> Thanks for pointing this out. While we mostly follow prior works like PiSSA [2] and MiLoRA [3] in the table format, we provide the parameter count and training time for commonsense reasoning tasks for LLaMA-3 and Falcon-Mamba below and will revise the tables in our paper.
>
> LLaMA3: (3epoch for lily)
>
> | method | param | training time | memory  |
> | ------ | ----- | ------------- | ------- |
> | LoRA   | 56M   | -             | -       |
> | PiSSA  | 83.8M | -             | -       |
> | MiLoRA | 56.6M | -             | -       |
> | Lily   | 1.2M  | 8.0627778h    | 23.2GiB |
>
> Falcon-Mamba: (1epoch for Lily and LoRA)
>
> | method               | param | training time | memory  |
> | -------------------- | ----- | ------------- | ------- |
> | LoRA                 | 3.7M  | 3.8275h       | 20.8GiB |
> | Lily ($\Delta$ + in) | 3.7M  | 3.975h        | 20.8GiB |
> | Lily (in)            | 3.3M  | 3.7833h       | 20.0GiB |
>
> We do not report the training time and memory of LoRA, PiSSA and MiLoRA in the LLaMA3 table because our results of LoRA, PiSSA and MiLoRA are taken from MiLoRA paper and the authors do not include those metrics (actually even without explicit parameter count). We are also constrained by our limited hardware resources in the discussion periord to reproduce them. We will provide them once we have proper resources to reproduce the results! We do have complete results for the Falcon-Mamba LLM experiment however.
>
> From the tabels, the first thing which we do not highlight in our paper is the shockingly low parameter count we used for Lily, despite beating all the compared methods (we will add this in our revised version). To give a convincing validation, we reproduce our results and the logs are below:
>
> >hellaswag:
>  test:10042/10042 | accuracy 9304  0.9265086636128261
>  boolq:
>  test:3270/3270 | accuracy 2385  0.7293577981651376
>  social_i_qa:
>  test:1954/1954 | accuracy 1520  0.7778915046059366
>  piqa:
>  test:1838/1838 | accuracy 1573  0.8558215451577802
>  winogrande:
>  test:1267/1267 | accuracy 1056  0.8334648776637726
>  ARC-Challenge:
>  test:1172/1172 | accuracy 909  0.7755972696245734
>  ARC-Easy:
>  test:2376/2376 | accuracy 2131  0.8968855218855218
>  openbookqa:
>  test:500/500 | accuracy 414  0.828
>
>
> Overall, we can observe that the **training time and memory usage of Lily is comparable to LoRA when using the same order of parameter**s, while having a notable advantage when used to adapt LLaMA3 in commonsense reasoning tasks using a much reduced parameter count.
>
> ---
>
> ## Q2(3): The experimental results do not include MNLI and QQP, despite these tasks being listed in the configuration table.
>
> This is indeed a point warrants explanation. We do not run all the tasks from GELU benchmark because all of our experiments are run in a single NVIDIA RTX 4090 GPU, which run extremely slowly on MNLI and QQP task and this makes it hard to perform some basic simple hyperparameters searching like searching the scaling factors $s$. For example, it takes roughly 30 minutes to run 1 epoch on MNLI dataset.
>
> In the configuration table we mark the $ne\_1$ and $ne\_2$ of MNLI and QQP as 0 to inform the readers that these two tasks are from GELU benchmark, but we do not actually adapt the model on these two tasks and report the results.
>
> **We promise to include these two tasks in a revised version of the paper once we have better computational resources.**
>
> [1] Yongchang Hao, et al. FLORA: Low-Rank Adapters Are Secretly Gradient Compressors
>
> [2] Fanxu Meng, et al. PiSSA: Principal Singular Values and Singular Vectors Adaptation of Large Language Models
>
> [3] Jingfan Zhang, et al. MiLoRA: Efficient Mixture of Low-Rank Adaptation for Large Language Models Fine-tuning
>
> ---
>
> If you still have concerns, please reply to us and we will try our best to address all of your concerns! We want to kindly restate that although our theoretical analysis is no longer used and we will not show them in the main text, replacing it with empirical analysis actually strengthen our claim :)

---

> ### Author Response · Authors · 2024-11-21
>
> Dear Reviewer F3Lx:
>
> We thank you again for your valuable feedback! Our responses were posted four days ago, but it appears that none of the reviewers have provided further thoughts or questions. It seems that you may have missed the notification about our responses. If that is the case, we would like to hear from you and are open to any further questions. We will address all of the concerns and improve our paper according to your feedback!

---

> ### Comment · Reviewer_F3Lx · 2024-11-23
>
> I appreciate the authors' clarifications, which have helped to make certain aspects of the work clearer during the rebuttal process.
>
> After carefully reviewing the responses, it is evident that Lily can be understood as a LoRA variant featuring deliberately designed layer-wise (local) and model-wise (global) shared low-rank matrices, effectively extending the spirit of MoE within the LoRA framework. However, I still have several critical concerns:
>
> 1. As stated, **"an adaptation target $W$, there's only one $A$ for it (whether it's shared in a range of layers or not) and multiple model-wide $B$s for it"**. Given such, I agree that double summation in Eq. 9 is not needed. In light of that, within the hierarchical framework where $B_{i}$ is applied in each layer, **the gradient analysis of $B_{i}$ then becomes problematic.** Without loss of generality, for the $k^{th}$ layer, $y^{k} = W^{k} y^{k-1} = (W_{0}^{k} + A_{0}^{k} \sum_{i=1}^{N_{e}} S_{i} B_{i}) y^{k-1} = (W_{0}^{k} + \Delta W^{k}) y^{k-1}$. By chain rule, $\frac{\partial{\mathcal{L}}}{\partial B_{i}} = \frac{\partial \mathcal{L}}{\partial \Delta W^{k}} \frac{\partial \Delta W^{k}}{\partial B_{i}} + \frac{\partial \mathcal{L}}{\partial y^{k-1}} \frac{\partial y^{k-1}}{\partial B_{i}}$ because $y^{k-1}$ is also a function of $B_{i}$. However, the gradients of $\frac{\partial \mathcal{L}}{\partial y^{k-1}} \frac{\partial y^{k-1}}{\partial B_{i}}$ are omitted in Eq. 12 for each layer, leading to errors in Eq. 13 and 14.
>
> 1. The authors acknowledge that **"we remove the theoretical analysis of the rank (section 3.1)"** and that **"current analysis is based on strong assumptions, as reviwer dXjD points out and lacks rigourous proof as you also points out, which is sadly beyond our current expertise"**. Given such, the motivation (e.g., gradient compressor) and methodology (Lily v1 and v2) require significant revision to align with the paper's claims and intended contributions.
>
> While I do appreciate the extensive empirical evaluations demonstrating the effectiveness of Lily, the concerns outlined above indicate that the work requires substantial revision that exceeds the scope of minor adjustments during the rebuttal phase. Hence, I am maintaining my original score.

---

> > ### Author Response · Authors · 2024-11-23
> >
> > Thank you for being the first reviewer to respond. One important thing to note is that our revised paper is uploaded and the **changes are in fact very minor, with no changes to the claim, core motivation and contributions.**
> >
> > Several things to clarify here:
> >
> > - After examining, we acknowledge that the Eq. 13 and Eq. 14 indeed have errors. Thank you for your patience. Additionally, as stated before, we remove the whole section completely from the paper. (please see our revised paper).
> > - The **central motivation is LoRA has low-rank weight update and information sources for a target is limited in a particular layer.** This is directly observed from LoRA, rather than provided by any work, nor does it has anything to do with the "gradient compressor" part.
> > - We want to kindly remind you that in the submitted version, **the gradient compressor analysis only serves as a theoretical introduction to why Lily has high-rank weight updates. It has nothing to do with our core motivation and contribution.** Also, now that we provide actual experiment, theoretical introduction is in fact not needed and do not undermine the claims or contributions.
> > - **The contribution is a novel PEFT method with inter-layer interaction and high-rank weight update. Again, it has nothing to do with the previous "gradient compressor" part and we provided extensive empirical analysis to demonstarte this.**
> > - We are confused when you say "methodology (Lily v1 and v2) require significant revision". First thing, v2 is not actually a version 2 or some major upgrades to Lily. **Basically, it is Lily**, and the so-called v1 is just a special case. Second, we wonder what you mean by "significant revision". In the revised paper, we do add **one sentense** to notify readers that number of LPs can be flexbly set and you [readers] can refer to the appendix for more information. **Meanwhile, we also want to note that the core innovation is the dynamics of HP and the routers, not how LPs are set. (i.e. , sharing LP is nothing new)**.
> > - There's no v2 or v1 of Lily, **the terms are for the convenience when replying to your concerns, not actual version 1 and version 2**. **There's just Lily.**
> >
> > To summarise, the core problem many reviewers (including you) point out is the "gradient compressor" part, whether it's presentation or correctness. However, we want to kindly remind you that **none of this paper's ctritical component (motivation, contribution and method) has anything to do with this part**. Basically in the submitted version, **it serves as an theoretical introduction to why Lily has high-rank weight updates** and removing it does essentailly no harm. To make up for its removal, we provide an empirical analysis instead.

---

> > ### Author Response · Authors · 2024-11-27
> >
> > Dear Reviewer F3Lx,
> >
> > Thank you once again for your constructive feedback. As the discussion period is drawing to a close, we would like to hear from you regarding our last response to you. In short, we argue that the revised paper does not contain major revisions that violate the central components of the paper (e.g., motivation, claim, etc.). We have also clarified some aspects of our method to help you better assess our work. We hope our response addresses your concerns and helps you evaluate our work.
> >
> > We look forward to your thoughts and hope to bring this review process to a successful conclusion.

---

> ### Author Response · Authors · 2024-12-01
> **Please reply to us and consider raising your score**
>
> Dear reviewer F3Lx,
>
> Thank you so much for your precious time and effort. As the discussion period is coming to its end, could you kindly read our responses and reply to us? If we have successfully addressed your concerns, we respectfully ask that you consider raising your score accordingly.
>
> Thank you again for your attention and effort!

---

### Official Review · Reviewer_HxWK · 2024-11-04

**Soundness:** 3
**Presentation:** 1
**Contribution:** 2
**Rating:** 5
**Confidence:** 2

**Summary:**

This paper proposes a novel hierarchical adaptor for fine-tuning the Transformer. The results show a high advance of proposed method on the NLU and computer vision tasks.

**Strengths:**

1. The proposed methodology that considers cross-layer interactions is new to me.
2. The empirical results show the method's potential in various domains.

**Weaknesses:**

The major flaws are about the presentation.

1. Some derivation steps could be merged and only the important steps can be given. The blanks could be reserved for discussing the intuitions and analysis.
2. The fonts in the figures are too small. Normally they should be larger than the smallest font in the main body.

**Questions:**

How does the proposed method affects the efficiency of the adaptation?

---

> ### Author Response · Authors · 2024-11-17
> **Replies to Reviewer HxWK**
>
> We greatly thank you for your constructive and helping feedback! We provide a detailed response to address all of your concerns below. An important thing to note is that we remove the theoretical analysis and leave it in the Appendix. **Instead, we provide a empirical analysis that reflects practical aspect of our method to validate our claim in our reply to your Q1, and we believe this makes much more convincing statements!** We will add this in our revised version as well :)
>
> ---
>
> ## Q1: Some derivation steps could be merged and only the important steps can be given. The blanks could be reserved for discussing the intuitions and analysis.
>
> Thanks for pointing that out. Following your advise and other reviwers' feedback, we directly remove the theoretical analysis part (section 3.1) in the main text and leave it to Appendix. Instead, we provide a numerical analysis of the **actual rank gain** using Lily below when using MRPC task after 20 epochs. For LoRA we use a hidden dimension of 8 and for Lily we use 32 and 8, respectively.
>
> | method      | param | layer1 | layer2 | layer3 |
> | ----------- | ----- | ------ | ------ | ------ |
> | LoRA  | 0.3M  | 27     | 29     | 24     |
> | Lily  | 0.3M  | 224    | 225    | 222    |
> | Lily  | 0.05M | 75     | 76     | 77     |
>
> It can be observed that the rank gain of using Lily is significantly higher than LoRA. **This strong experimental and empirical results validate our claim, despite we no longer rely on theoretical derivation.**
>
> Meanwhile, the intuitions and multiple analysis are provided in the appendix (e.g. appendix A, C, D, E, F, G, H, I, J).
>
> ---
> ## Q2: The fonts in the figures are too small. Normally they should be larger than the smallest font in the main body.
>
> Thanks for your feedback, we will change the fonts size accrodingly to have a better presentation in our revised version!
>
> ---
> ## Q3: How does the proposed method affects the efficiency of the adaptation?
>
> Obviously the proposed method introduce a certain level (despite small) of complexity. However, the speed and memory comsumption are comparable to those of LoRA's. We provide a speed and memory analysis down below running COLA task from GLUE, comparing LoRA, Lily and FFT using RoBERTa-Base. We run 10 epochs with batchsize 64 to test the hardware efficiency.
>
> | method      | param | run time (sec)       | memory      |
> | ----------- | ----- | -------------------- | ----------- |
> | LoRA        | 0.3M  | 35.0998492240905     | 3040MiB     |
> | Lily        | 0.3M  | 38.60865545272827    | 3608MiB     |
> | Lily (mono) | 0.3M  | **34.8127372264862** | **1778MiB** |
> | FFT         | 125M  | 56.8776972293853     | 4353MiB     |
>
> It can be observed that typical Lily outperform LoRA by a notable margin in our experiments with **comparable hardware efficiency and speed**. **Additionally, using the monoscale version of Lily (Lily mono) introduced in the paper appendix could further achieve hardware efficiency that greatly surpass LoRA.** Meanwhile, the gap of the speed and memory between LoRA and Lily is vey small, making it neglible in the most common use case of PEFT (i.e. smaller-scale models and small-scale dataset for domain-specific fine-tuning). **Overall, the proposed method performs better with comparable hardware efficiency of typical PEFT methods.**
>
> ---
>
> If you still have concerns, please reply to us and we will try our best to address all of your concerns! We want to kindly restate that although our theoretical analysis is no longer used and we will not show them in the main text, replacing it with empirical analysis actually strengthen our claim :)

---

> ### Author Response · Authors · 2024-11-21
>
> Dear Reviewer HxWK:
>
> We thank you again for your valuable feedback! Our responses were posted four days ago, but it appears that none of the reviewers have provided further thoughts or questions. It seems that you may have missed the notification about our responses. If that is the case, we would like to hear from you and are open to any further questions. We will address all of the concerns and improve our paper according to your feedback!

---

> > ### Author Response · Authors · 2024-11-27
> >
> > Dear Reviewer HxWK,
> >
> > Thank you once again for your constructive feedback. As the discussion period is drawing to a close, we would appreciate hearing from you regarding our responses. We have revised the paper according to your feedback, for example, by enlarging the fonts in the figures, and we hope your concerns have been properly addressed.
> >
> > We look forward to your thoughts and hope to bring this review process to a successful conclusion.

---

> ### Author Response · Authors · 2024-12-01
> **Please reply to us and consider raising your score**
>
> Dear reviewer HxWK,
>
> Thank you so much for your precious time and effort. As the discussion period is coming to its end, could you kindly read our responses and reply to us? If we have successfully addressed your concerns, we respectfully ask that you consider raising your score accordingly.
>
> Thank you again for your attention and effort!

---

### Official Review · Reviewer_Njua · 2024-11-04

**Soundness:** 3
**Presentation:** 3
**Contribution:** 2
**Rating:** 6
**Confidence:** 4

**Summary:**

In this paper, the authors propose a parameter-efficient fine-tuning method based on the concept of LoRA. Specifically, they introduce a technique to construct the B matrix through a weighted combination of inter-layer weights during inference. This approach allows the adapter to overcome the low-rank limitation, improving its expressiveness. The proposed method is motivated by findings that suggest LoRA functions as a gradient projector, with updates primarily influenced by the initialization of the A matrix. The authors evaluate their method on both transformer-based and Mamba-based models, comparing it with baseline methods across various NLP and image tasks. Additionally, they conduct ablation studies and qualitatively assess the learned features.

**Strengths:**

- Overall, this paper is well-written. The authors provide a comprehensive description of the proposed method, and the appendix is detailed.

- The authors apply the proposed method to various models across different modalities, demonstrating its effectiveness. They include both quantitative and qualitative evaluations.

- The proposed method is inspired by a thorough analysis of the existing LoRA approach.

**Weaknesses:**

- Since the authors derived the forward path of the proposed method in Section 3.2, would it be possible to analyze the rank gain? A theoretical analysis of the improvement assuming a single layer could be helpful.

- It would be beneficial if the authors could further evaluate the improvement qualitatively, as the current gains are not particularly substantial. For example, in the RoBERTa-base experiments, the proposed Lily method only outperforms others on 2 out of all tasks. Additionally, AdaLoRA does not outperform standard LoRA, which is somewhat counterintuitive. Could the authors explore conducting experiments on larger BERT models? Given that different PEFT methods involve varying numbers of learnable parameters, is it fair to compare these methods directly in light of these parameter differences?

- The authors have renamed the A and B matrices in LoRA as the downward low-dimensional projector and upward high-dimensional projector, respectively.

**Questions:**

- The authors might want to discuss the difference of the proposed method with several related recent works. For example, in [HydraLoRA], the authors introduced a LoRA MoE method which share the same A matrices but also a mixture of B matrices. In [LoRA Asymmetry], the authors proposed freezing A matrices and only training B, backed by a another line of theoretical study. Another related works, [LoRA+], the authors analyze the gradient of LoRA matrices and propose to use a larger learning rate for B to improve the performances.

[Tian, Chunlin, et al. "HydraLoRA: An Asymmetric LoRA Architecture for Efficient Fine-Tuning." arXiv preprint arXiv:2404.19245 (2024).]

[Zhu, Jiacheng, et al. "Asymmetry in low-rank adapters of foundation models." arXiv preprint arXiv:2402.16842 (2024).]

[Hayou, Soufiane, Nikhil Ghosh, and Bin Yu. "Lora+: Efficient low rank adaptation of large models." arXiv preprint arXiv:2402.12354 (2024).]

---

> ### Author Response · Authors · 2024-11-17
> **Replies to Reviewer Njua: Part 1**
>
> We greatly appreciate your insightful and constructive feedback! We provide a detailed response to all of your concerns below. An important thing to note is that since the current theoretical analysis has strong assumptions and insufficient proof as pointed out by other reviwers, we remove them in the main text and leave them in the Appendix instead. **However, we validate our claim using a stronger and more convincing empirical analysis in our reply to your Q1 based on real-world experiments and we think this actually make our claim and results much more convincing :)**
>
> ---
> ## Q1: A theoretical analysis of the improvement (rank gain) assuming a single layer could be helpful.
> Thanks for pointing that out. Here is an analysis from both theoretical and emprical aspect **and we will add it to our revised version (the empirical one goes to the main text and the theoretical one goes to appendix for readers who are interested)**.
>
> ### Theoretical:
> The formulas we provided generally indicate a much higher upper-bound for the rank of the weight updates ($\Delta W$).
>
> **When using LoRA**, the weight updates is illustrated in Eq. 8 in the paper $W = W_0 - \eta \sum_{t=0}^{T}\big[ A_0A_0^T(\nabla_{W}\mathcal{L}_t)\big]$ .
>
> Therefore the weight updates is dominated by the term $A_0A_0^T(\nabla_{W}\mathcal{L})$. Since it's a general formula of some matrices, we can calculate the upper-bound of its component by basic rules: $rank(A_0A_0^T)\leq min(rank(A_0),rank(A_0^T))=d$,$rank(\nabla_{W}\mathcal{L}) \leq min(C_{in} \times C_{out})=C_{in}$ (here we assume $C_{in}=C_{out}$). Therefore $rank(A_0A_0^T(\nabla_{W}\mathcal{L})) \leq min(C_{in}, d) = d$. Since $d$ in LoRA is a much smaller hidden dimension (say 32, 16 or even 8 and 4), the weight updates are essentialy limited by this low upper-bound of the rank.
>
> **When using Lily**, the weight updates is in Eq. 14 $W = W_0 - \eta \sum_{t=0}^{T}\Big[ \sum_{i=1}^{N_e} S_i\big[\sum_{j=1}^{N_l}{C_{i,j}A_0 A_0^j}^T(\nabla_{W_j}\mathcal{L}_t)\big]\Big]$.
>
> Therefore the weight updates is actually dominated by the term $\sum_{i=1}^{N_e} S_i\big[\sum_{j=1}^{N_l}{C_{i,j}A_0 A_0^j}^T(\nabla_{W_j}\mathcal{L}_t)\big]$.
>
> Although $rank(A_0 {A_0^j}^T(\nabla_{W_j}\mathcal{L}_t)) \leq d$,
>
> due to the **different $A_0^j$ and varying $C_{i,j}$**, the overall summation of $\sum_{j=1}^{N_l}{C_{i,j}A_0 A_0^j}^T(\nabla_{W_j}\mathcal{L}_t)$
>
> has a rank upper-bound $N_l \times d$. Moreover, the summation of the summation (i.e. $\sum_{i=1}^{N_e} S_i$) with **varying $S_i$** further leads to a overall rank upper-bound of $N_e \times N_l \times d$.
>
> In practice, if we take $N_e$ as 6 and $N_l$ as 10, the overall upper-bound of Lily and LoRA is $60 \times d$ versus $d$. If the same upper-bound for LoRA is to be achieved, a 60x larger $d$ is typically required and it will dramatically hinder the efficiency of PEFT. In essence, we provide much higher upper-bound for the weight updates despite employing a low rank model strcture.
>
> ### Empirical:
>  We provide an emprical analysis of the actual rank gain of Lily versus LoRA. Here we use MRPC dataset from GLUE as the dataset and we report the **actual rank** of the weight updates during a single forward pass of layers 1, 2 and 3 down below. For LoRA we use a hidden dimension of 8 and for Lily we use 32 and 8, respectively. The target is $W_q$ but the resuls generalize well to other targets as well. We run 20 epochs for the analysis. Note that we only use 2LPs and 2HPs in Lily to match the parameter count.
>
> | method      | param | layer1 | layer2 | layer3 |
> | ----------- | ----- | ------ | ------ | ------ |
> | LoRA  | 0.3M  | 27     | 29     | 24     |
> | Lily | 0.3M  | 224    | 225    | 222    |
> | Lily  | 0.05M | 75     | 76     | 77     |
>
> As the table shows, the rank gain of Lily is substantial compared to LoRA when using the same order of parameters. Meanwhile, Lily still outperforms LoRA in terms of the high-rank weight update when using only 0.05M parameters, which is notable. **This empirical analysis (we will add more datasets in our revised version of the paper) demonstrate our claim that Lily achieves high-rank weight updates despite using a low-rank structure, and it is actually much more convincing than our current theoretical analysis!**

---

> ### Author Response · Authors · 2024-11-17
> **Replies to Reviewer Njua: Part 2**
>
> ## Q2: The current gains are not particularly substantial.
> That's because the hyper-parameters setting in the paper is significantly simple without any specific tuning (i.e. we mostly fix the learning rate and only search the scaling factor from a small range {0.01, 0.1}. To make this statement more convincing, we further tune some hyperparameters for certain NLU tasks and here are the results:
>
> |       | original | tuned    |
> | ----- | -------- | -------- |
> | SST-2 | 95.0     | **95.1** |
> | QNLI  | 92.5     | **93.0** |
> | STS-B | 90.8     | **91.0** |
> | COLA  | 66.0     | **66.2** |
> | STS-B | 90.8     | **91.0** |
> | RTE   | 81.6     | 81.6     |
>
> The overall performance now is 86.2, which is $1$% higher than FFT and is quite substantial considering our parameter efficiency. It is important to note that we are still using simple searching, but with a wider range. Meanwhile, performance of Lily in vision tasks (VTAB-1K in our paper) is more pronounced than in NLU (achieving a notable margin arcoss 19 tasks and uses the least amount of parameters), as Lily's design will intuitively enable a more comprehensive and expressive visual adaptaton with its cross-layer connection.
>
> ---
> ## Q3: AdaLoRA does not outperform standard LoRA, which is somewhat counterintuitive.
> That's a good point. AdaLoRA indeed proposes an adaptive budget allocation strategy to make the most usage of the parameters budget. However, we believe that while intuitively it will make AdaLoRA more powerful, the **outcome is subject to all kinds of factors involved** in the adaptation (**hyperparameters** for example). Additionally, FFT actually also underperforms LoRA and BitFit in the RoBERTa-Base experiment, which further illustrates our point that there're factors like the size of the dataset or model, hyperparameters or the category of the task that can influence the overall outcome.
>
> ---
> ## Q4: Could the authors explore conducting experiments on larger BERT models?
> We further conduct experiments on RoBERTa-Large and results are provided below. **We will also include this in our revised version**.
>
> |      | params   | COLA     | MRPC     | QNLI     | RTE      | SST-2    | STS-B    | Ave      |
> | ---- | -------- | -------- | -------- | -------- | -------- | -------- | -------- | -------- |
> | Lily | **0.5M** | **68.4** | **90.9** | **94.8** | **88.4** | 95.6     | 91.9     | **88.4** |
> | LoRA | 0.8M     | 68.2     | 90.2     | **94.8** | 85.2     | 96.2     | 92.3     | 87.8     |
> | FFT  | 356M     | 68.0     | 90.9     | 94.7     | 86.6     | **96.4** | **92.4** | 88.2     |
>
> It can be observed that the advantage of Lily is still notable when adapting larger RoBERTa models in natural language understanding tasks. Meanwhile, Lily achieves the best performance in 4 out of 6 tasks using only $60$% of the parameters of LoRA, all the while with minimum hyperparameters tuning (we fix the learning rate and perform simple scaling factor searching as before). This demonstrates Lily's advantage of being model-size agnostic and tuning-friendly (i.e. good performance with no complex tricks applied.)
>
> ---
> ## Q5: Is it fair to compare these methods directly in light of these parameter differences?
>
> We basically follow prior works in PEFT, which compare the methods using varying but still comparable (in the same order) parameters count, along with the actual adaptation performance. **We believe that** **as long as the proposed method uses the least amount of parameters and outperform all other methods**, the comparison is fair and convincing.
>
> ---
> ## Q6: The authors have renamed the A and B matrices in LoRA as the downward low-dimensional projector and upward high-dimensional projector, respectively.
>
> We believe that our renaming actually has a convincing reason: the adapters in methods like LoRA, AdaLoRA, HydraLoRA, etc, are specific to an adaptation target (say, a matrix $W$). For example, the $A$ and $B$ in LoRA is used to adapting only one specific target. While in Lily, the adapters are either model-wide or shared among a few neighboring layers. To distinguish the semantics, we choose to name the adapters based on their operation on the dimensionality (e.g., those that reduce the dimension are named downward low-dimensional projectors and vice versa).
>
> If you still have strong concern about the naming issue, we would consider changing it in the revised version.

---

> ### Author Response · Authors · 2024-11-17
> **Replies to Reviewer Njua: Part 3**
>
> ## Q7: Discuss the difference of the proposed method with several related recent works. HydraLoRA, LoRA Asymmetry, LoRA+
>
> Thanks a lot for pointing out these great and insightful works! HydraLoRA is a highly related work and has a similar design of asymmetric architecture, the reason we didn't cite it is that its acceptance to NeurIPS is later than our submission to ICLR2025. We discuss these method below and will add them in our revised version.
>
> - HydraLoRA: This is an interesting work that uses a similar asymmetric design (i.e. shared A with multiple Bs). We believe that our work, along with works like HydraLoRA [1] and MosLoRA [2] are **concurrent** pioneers of introducing MoE inside the typical LoRA adapters. While the design has similarity, the foundamental difference is that Lily (ours) deploy this asymmetric architecture **model-widely** (i.e. across the layers as our title states) and this inherently enable communications among different layers, which enables **feature merging** (please refer to section4.5) and **allows a more comprehensive visual adaptation**. Meanwhile, our work also differs in the **diversity of our experiments** (diffusion, commonsense reasoning and visual adaptation) and the exploration of newly introduced but not widely used mamba-based models.
> - LoRA Asymmetry: Another insightful work. The core idea is also properly demonstarted in Flora [3] (i.e. It is the matrix B that dominates the overall adaptation) and this was also the premise of our work. (**However, we opt for an empirical analysis now**) The main difference is that it does not explicitly introduce multiple Bs for an adaptation (which Lily and HydraLoRA do) and this approach could lack expressiveness as it does not change the structure of LoRA. **However, this work could be combined with our work by making the As frozen and only tuning all the Bs to further boost parameter efficiency and the overall performance.**
> - LoRA+: Also a great work that deeply explores the dynamics of LoRA. This work also offers an important insight that the learning rate does not to be the same in A and B, since B is more important for the adaptation. **This idea can be employed in Lily as well to further boost the performance and expressiveness of the adaptation**.
>
> [1] Tian, Chunlin, et al. HydraLoRA: An Asymmetric LoRA Architecture for Efficient Fine-Tuning
>
> [2] Taiqiang Wu, et al. Mixture-of-Subspaces in Low-Rank Adaptation.
>
> [3] Yongchang Hao, et al. FLORA: Low-Rank Adapters Are Secretly Gradient Compressors
>
> ---
>
> If you still have concerns, please reply to us and we will try our best to address all of your concerns! We want to kindly restate that although our theoretical analysis is no longer used and we will not show them in the main text, replacing it with a more convincing empirical analysis actually strengthen our claim :)

---

> ### Author Response · Authors · 2024-11-21
>
> Dear Reviewer Njua:
>
> We thank you again for your valuable feedback! Our responses were posted four days ago, but it appears that none of the reviewers have provided further thoughts or questions. It seems that you may have missed the notification about our responses. If that is the case, we would like to hear from you and are open to any further questions. We will address all of the concerns and improve our paper according to your feedback!

---

> > ### Comment · Reviewer_Njua · 2024-11-27
> > **Response**
> >
> > I would like to thank the authors for providing additional experimental results and discussions. Regarding the naming of model matrices, as a reviewer, I do not have strong preferences, as I understand the role of the proposed projectors. However, I suggest considering a renaming or adding further clarification, as this might make your method more easily adoptable, given the popularity of the existing LoRA structure.
> > Overall, I maintain my positive assessment of this work.

---

> > > ### Author Response · Authors · 2024-11-27
> > >
> > > Thank you for your insightful feedback and positive assessment of our work! We will clarify the naming in our final version of the paper. Additioanlly, as you point out the popularity of the existing LoRA structure, we will seriously consider changing the variable names in our code implementation to make Lily more adoptable. Thank you again for your precious time and effort!

---

### Author Response · Authors · 2024-11-20
**Validate Our Claim and Invite Reviewers to Discussion**

We thank again for all the reviewers' insightful and helpful feedback!  To summarize, the reviewers highlight the strengths of our paper, including the **novelty of inter-layer interaction, reasonable motivation, comprehensive numerical analysis across multiple scenarios, and the generally promising potential of our proposed method.**

However, the reviewers also point out that the current theoretical analysis (Section 3.1) has several presentation issues, lacks sufficient and rigorous proof, and relies on some strong assumptions based on prior works. This makes our claim that "Lily achieves weight updates with higher rank than LoRA" less convincing from a theoretical perspective.

We do acknowledge that this analysis lacks sufficient proof and is based on some strong assumptions. Therefore, we decide to remove this section from the main text due to its current immaturity. **However, we actually test the rank of the weight updates of Lily and LoRA in multiple real-world experiments, validating our claim that "Lily achieves weight updates with higher rank than LoRA" from a much more convincing practical and empirical perspective. We believe that the removal of immature theoretical analysis does not undermine our claim and the empirical analysis, combined with our existing experimental results, further demonstrates the advantage of our proposed method.**

Here we provide the empirical rank analysis of Lily and LoRA. To be specific, we use the MRPC, COLA, SST-2 and STS-B tasks from the NLU experiments and test the rank of the $W_q$ when running 20 epochs for the former three tasks and 3 epochs for STS-B. The results below generalize well to other targets and layers as well. For LoRA we use a hidden dimension of 8 and for Lily we use 32 and 8, respectively. We use only a small number of LPs and HPs (2 or 3) in the experiment for simplicity and to make our claim more convincing. The weight updates here are of shape [768, 768]. **We will turn these analyses into figures and include them in a section in the main text to provide strong empirical evidence of our claim.**

MRPC:

| method | param | layer1 | layer2 | layer3 |
| ------ | ----- | ------ | ------ | ------ |
| LoRA   | 0.3M  | 27     | 29     | 24     |
| Lily   | 0.3M  | 224    | 225    | 222    |
| Lily   | 0.05M | 75     | 76     | 77     |

COLA:

| method | param | layer1 | layer2 | layer3 |
| ------ | ----- | ------ | ------ | ------ |
| LoRA   | 0.3M  | 56     | 34     | 37     |
| Lily   | 0.3M  | 313    | 312    | 324    |
| Lily   | 0.05M | 105    | 101    | 101    |

STS-B:

| method | param | layer1 | layer2 | layer3 |
| ------ | ----- | ------ | ------ | ------ |
| LoRA   | 0.3M  | 32     | 29     | 28     |
| Lily   | 0.3M  | 214    | 228    | 194    |
| Lily   | 0.05M | 151    | 148    | 151    |

SST-2 (3 epoch)

| method | param | layer1 | layer2 | layer3 |
| ------ | ----- | ------ | ------ | ------ |
| LoRA   | 0.3M  | 50     | 54     | 40     |
| Lily   | 0.3M  | 363    | 355    | 351    |
| Lily   | 0.05M | 109    | 115    | 114    |

**Additionally, we want to kindly invite all the reviewers to participate in the discussion. We provide detailed responses to all the concerns and we are determined to address all of them!**

---

### Author Response · Authors · 2024-11-23
**revised paper is out and we want to hear from you**

Dear reviewers and area chairs:

Thank you for your precious time and effort put into this work. We upload the revised version of the paper. One important thing to note is that changes are minor, with no changes to the central claim, motivation, experiment, method and contributions.

To summarise:
- Removal of the theoretical analysis and statements regarding the "gradient compressor" terminology. Key thing is that **originally, this part is trying to illustate why Lily has high-rank weight updates. Now with the inclusion of actual experiment testing the rank, we directly remove it from the paper.**
- Reminder that LPs for Lily can be flexbly set. (**one sentense in green**).
- RoBERTa-Large experiment (caption with blue text highlighted)
- Actual analysis of the rank for weight updates. (blue text).
- Hardware efficiency. (runtime or GPU memory consumption, blue text)
- Bigger font size for the figures.
- New citation to Hydra-LoRA (red text).

Again, the changes are minor and we want to restate that **removing the theoretical introduction changes nothing about this paper's central claim, motivation, experiment, method and contributions.** Thank you again for reviewing this paper. If you have further concerns, please respond and we will adress them.

---

### Author Response · Authors · 2024-11-24
**Examining Our Paper's Components to Identify Whether Major Revisions Exist**

Dear reviewers and area chairs,

We examine components of the paper to help you identify whether major revisions of the paper exist. (TL;DR: major revisions do not exist in our revision)

### Motivation 1: LoRA has low-rank weight updates.

 Previously this part is **introduced** with the "gradient compressor" part. However, **removing it in the revised version does not change the fact that LoRA has low-rank weight updates, because this is not a conclusion of any other papers but rather directly observed.** **This is not major revision bacause the motivation is essentially the same (LoRA has low-rank weight updates), but without the previous introduction.**

### Motivation 2: Information for LoRA target is limited in a particular layer.
This is also a **fact that does not depends on any works**, which can be observed in LoRA. (i.e., A and B are in a particular layer in LoRA). **This motivation leads to our inter-layer connection design.** We previously dicussed this in section 4.5 and now restate this in introduction. **This is also not major revision because this is already discussed in the paper and we simply restate it multiple times in the introduction.**

### Method: **No revision at all**.

One sentense is added to ask readers to read appendix to know how LPs can be set. This changes nothing, bacuase the appendix is also the same without any changes. **We kindly remind you that we never state that number of LPs should be the same as layer numbers and how to set LPs is not the core innovation of this paper, rather it's the HP and Routers.** ***Also, we already inform the readers how LPs are set in the appendix (submitted version).***

### Experiments:
  - Add actual rank analysis, this serves as analysis of why Lily has high-rank weight updates. **This is not major revision bacause this is a supplementary analysis.**
  - Bigger fonts of the figures. **This is not major revision because the content of the figures is the same, but with bigger fonts.**
  - RoBERTa-Large experiment. This is a supplementary experiment. **Zero new claims associated with it is proposed.**
  - Hardware efficiency analysis. **Also supplementary analysis to help readers understand the runtime and memory.**

### Claims and Contributions:
  - Lily has high-rank weight updates. **No revision at all**. Demonstrated by rank analysis.
  - Cross-layer interaction. **No revision at all**. This is discussed in section 4.5
  - Extensive experiments are conducted across various modalities, architectures, and model sizes. **No critical experiments are added.**

In general, the revision is mainly rephrasing the motivation and removing section 3.1. None of these changes the central component of the submitted paper. Regarding section 3.1, it is originally introduced to illustrate why Lily has high-rank weight updates and **has nothing to do with our contributions**. Now with the empirical analysis, this part is in fact not needed (plus many reviewers point out its incorrectness). **The conclusion of this post is that major revisions do not exist in our revision.**

---

### Author Response · Authors · 2024-12-02
**Dear reviewer HxWK, F3Lx and dXjD, please respond or consider raising your score**

Dear reviewer HxWK, F3Lx and dXjD,

Thank you for your precious time and effort. In the last day of the discussion, we kindly and finally ask you to respond to us. If we have addressed your concerns, please raise your score! Thank you again for your attention.

Authors of submission #3759

---

### Author Response · Authors · 2024-12-03
**Summary of Discussion Phase**

Dear Reviewers and Area Chairs,

We would like to thank you for your efforts. Below, we provide a summary of this discussion phase.

The initial review of this paper pointed out **several merits, including reasonable motivation (reviewer F3Lx), a novel approach (reviewer HxWK), comprehensive experiments (reviewers Njua and dXjD), and, in general, a promising contribution.**

The main weakness many reviewers pointed out is the presentation, particularly referring to the theoretical analysis of why Lily has high-rank weight updates. We acknowledge the immaturity of our analysis and have replaced it with a stronger empirical analysis. We believe this does not count as a major revision because:
1, **The high-rank weight updates are demonstrated by real-world experiments.**
2, The previous theoretical analysis did not contribute significantly to the overall contribution of our paper.

In general, the paper proposes a promising PEFT method with great performance, and its original weakness was mainly about its presentation. However, **after minor revisions, the presentation is now clear and well-written. Therefore, we have successfully addressed the concerns raised by the reviewers.**

---

### Meta-Review · Area_Chair_JsXj · 2024-12-19

**Metareview:**

This paper proposes low-rank
interconnected adaptation across layers to handle the existing restriction of LoRA. After the response, it receives mixed ratings, including three rejects, and one accept. The advantages, including the various combinations of models and the interesting idea of cross-layer interaction, are recognized by the reviewers. However, they are also concerned about the presentation of the methodology, unsatisfying experimental results, lack of validity of the assumptions, etc.

Though the authors make great efforts to provide detailed revision, two reviewers think the work requires substantial revision that exceeds the scope of minor adjustments during the rebuttal phase. And the main concerns are not well addressed in the response. After carefully reading the paper and reviews, I agree with them. For the revised paper, the full METHODOLOGY part is less than one page, which can support the above opinion from my point of view.

In summary, I also think the current manuscript does not meet the requirements of this top conference. I suggest the authors carefully revise the paper and submit it to another relevant venue.

**Additional Comments On Reviewer Discussion:**

All reviewers keep their original ratings unchanged. For the concerns of assumption, the authors try to move it from the paper. Two reviewers think the storyline is changed, and the paper needs substantial revision that exceeds the scope of minor adjustments during the rebuttal phase. I agree with them and think the current manuscript does not meet the requirements of this top conference.

---

### Decision · Program_Chairs · 2025-01-22

Reject